# Cellular energy regulates mRNA degradation in a codon-specific manner

Pedro Tomaz da Silva [1,2], Yujie Zhang[3], Evangelos Theodorakis[1], Laura D Martens[1,4], Vicente A Yépez[1], Vicent Pelechano [3] & Julien Gagneur [1,4,5]✉

## Abstract

**Codon optimality is a major determinant of mRNA translation and degradation rates. However, whether and through which mechanisms its effects are regulated remains poorly understood. Here we show that codon optimality associates with up to 2-fold change in mRNA stability variations between human tissues, and that its effect is attenuated in tissues with high energy metabolism and amplifies with age. Mathematical modeling and perturbation data through oxygen deprivation and ATP synthesis inhibition reveal that cellular energy variations non-uniformly alter the effect of codon usage. This new mode of codon effect regulation, independent of tRNA regulation, provides a fundamental mechanistic link between cellular energy metabolism and eukaryotic gene expression.**

**Keywords** mRNA Stability; Cellular Energy Metabolism; Tissue-specific Regulation; Codon Usage Bias; Codon Optimality-mediated mRNA Degradation
**Subject Categories** RNA Biology; Translation & Protein Quality

## Introduction

Codons encode in three nucleotides 20 amino acids in a redundant way. Importantly, codons coding for the same amino acid, or synonymous codons, are not functionally equivalent. In particular, codons differ on their optimality for being decoded by the translation machinery, which not only affects the rate of protein production but also the rate of messenger RNA (mRNA) degradation (Hoekema et al, 1987), via a pathway termed codon optimality-mediated mRNA degradation (COMD) (Bae and Coller, 2022). Mechanistically, it has been shown in yeast that a ribosome dwelling at a given non-optimal codon not only delays its translation but can also be recognized by the Ccr4-Not complex triggering mRNA degradation (Buschauer et al, 2020).

Differences in optimality between codons have been suggested to be mostly determined by variation in cognate tRNA concentration (reviewed in Hanson and Coller (2018)). In light of such an explanation, regulation of the tRNA pool composition between cell types and conditions could differentially affect mRNA translation and degradation. Consistent with this hypothesis, associations between codon usage and differential expression between tissues and conditions have been reported across eukaryotes (Gingold et al, 2014; Burow et al, 2018a; Guimaraes et al, 2020; Hernandez-Alias et al, 2020; Allen et al, 2022). However, whether regulation of the tRNA pool is generally causing these associations is debated. While variations of the tRNA pool have been reported by some studies (Gingold et al, 2014; Goodarzi et al, 2016; Hernandez-Alias et al, 2020), other studies, including some with advanced tRNA sequencing protocols (Schmitt et al, 2014; Pinkard et al, 2020; Behrens et al, 2021), reported surprisingly stable proportions of the tRNA pool per anticodon. Hence, under which conditions and how codon optimality plays a role in differential gene expression remains unclear.

## Results

Here we analyzed the effect of codon optimality on differential gene expression by first looking at mRNA stability. To this end, we considered changes in the ratio between exonic and intronic RNA-Seq read coverage as a proxy for the variation of mRNA stability ((Gaidatzis et al, 2015), Methods). We processed 7771 RNA-Seq human post-mortem samples from the GTEx project spanning 49 tissues and 528 individuals (Fig. 1A; Appendix Fig. S1). Typically, mRNA stability varied 2.7-fold between tissues (median fold change between lowest and highest decile). In comparison, mRNA abundance typically varied by 4.2-fold between tissues (transcript per million median fold-change between lowest and highest decile). This observation shows that mRNA stability variations substantially contribute to between-tissue mRNA regulation and underscores the importance of post-transcriptional processes in gene regulation, in agreement with previous studies (Rabani et al, 2011; Duan et al, 2013).

As a first analysis, we aggregated all GTEx samples into tissues in order to focus on tissue-specific rather than individual-specific variations. To study mRNA stability regulation, we defined the relative half-life as the ratio of the mRNA stability in a tissue to its

[1]School of Computation, Information and Technology, Technical University of Munich, Garching, Germany. [2]Munich Center for Machine Learning, Munich, Germany. [3]Scilifelab, Department of Microbiology, Tumor and Cell Biology, Karolinska Institutet, Stockholm, Sweden. [4]Computational Health Center, Helmholtz Center Munich, Neuherberg, Germany. [5]Institute of Human Genetics, School of Medicine, Technical University of Munich, Munich, Germany. ✉E-mail: gagneur@in.tum.de

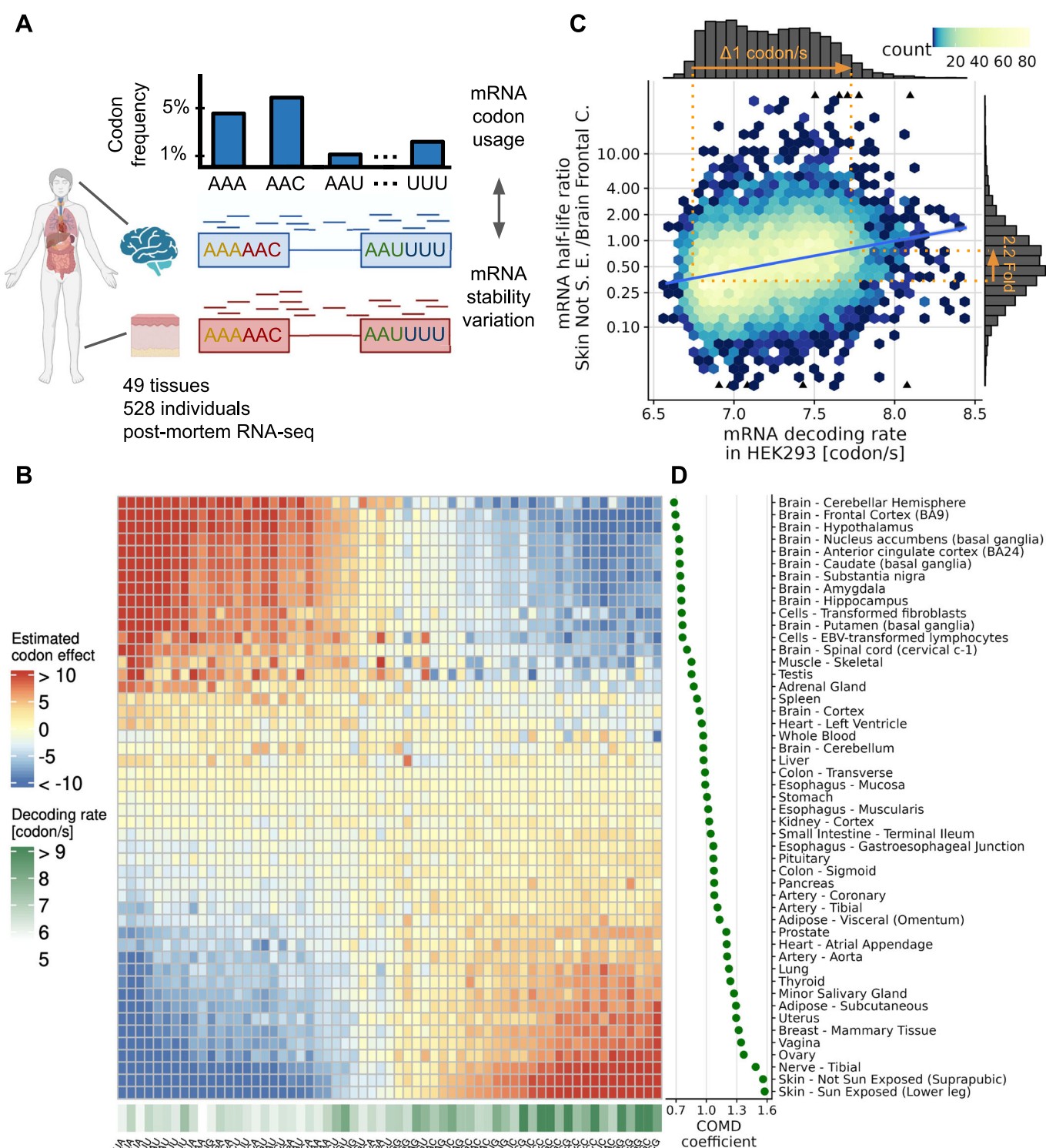

**Figure 1. Codon optimality associates with mRNA stability variations between tissues.**

(**A**) Variations in mRNA stability for every gene are estimated using the ratio of the number of exonic reads, mostly reflecting the balance between mRNA synthesis and degradation, to the number of intronic reads, mostly reflecting mRNA synthesis (Gaidatzis et al, 2015). We then investigated how those estimated variations in mRNA stability associate with codon usage. (**B**) Association (regression slope, Appendix Fig. S2) between codon frequency (column) and relative mRNA half-life for each tissue (rows), along with codon decoding rate measured in the HEK293 cell line (lower track), a measure of codon optimality. The tissues are ordered by increasing COMD coefficient (right panel, Methods). (**C**) mRNA half-life ratio of suprapubic skin to frontal cortex against mRNA decoding rate. The mRNA decoding rates are expressed in codons per second and computed using codon decoding rates measured in HEK293 cells (Methods). According to the linear regression (blue line), a decoding rate change of 1 codon per second in HEK293 associates with 2.2-fold larger mRNA stability ratios (orange annotations). (**D**) COMD coefficient across tissues.

mean across tissues (Methods). We observed strong associations between codon frequencies and tissue-specific relative half-life (Fig. 1B). Strikingly, in each tissue, the associations of codons with mRNA half-life ranked according to various codon optimality measures (Fig. 1B; Appendix Fig. S3). This suggested a global modulation of codon optimality effects on mRNA half-life rather than a regulation at the level of individual codons. We next asked whether these codon-level associations were also reflected at the mRNA level. To this end, we considered for each mRNA its decoding rate in HEK293 cells as a codon optimality measure (Dana and Tuller, 2015). Importantly, we did not assume that decoding rates are conserved across tissues. Instead, we used the HEK293 decoding rates as a measure of codon optimality and investigated how this codon optimality measure associates with relative mRNA half-life. The dynamic range of HEK293 mRNA decoding rates spanned about 1 codon per second (difference between the slower and the faster 4th percentile, Fig. 1C). Remarkably, variation in this range was associated with changes in half-life of surprisingly large amplitudes, including a 2.2-fold-change between suprapubic skin and frontal cortex (Fig. 1C). These results suggest that the amplitude of the effect of codon optimality on mRNA degradation is modulated and is an important contributor to mRNA regulation across human tissues.

To systematically study variations of codon optimality effects on mRNA half-life, we introduced a new metric termed codon optimality-mediated mRNA degradation (COMD) coefficient. The COMD coefficient quantifies how much the relative half-life of an mRNA is predicted to change given a decoding rate increase of 1 codon per second in HEK293. The higher the COMD coefficient in a given sample, the more beneficial the usage of optimal codons for mRNA stability. The COMD coefficient was maximal in sun-exposed skin, reaching a value of 1.6-fold-change/codon/s (Fig. 1D). Assuming causality, this suggested that recoding an mRNA such that its decoding rate is 1 codon per second faster in HEK293, could increase its half-life by 1.6-fold in sun-exposed skin relative to the average tissue. The COMD coefficient was minimal in several brain tissues with values of about 0.7-fold-change/codon/s. Notably, the ranking of the tissue COMD coefficients was similar when using decoding rates reported in cell lines other than HEK293 (Appendix Fig. S4), showing that these observations were qualitatively robust to the exact decoding rates used as reference for defining the COMD coefficient.

We next asked which biological processes, if any, could cause this apparent modulation of the effect of codon optimality on mRNA degradation. Using the Gene Ontology, we found the strongest associations for mitochondrial ATP synthesis pathways, the expression of whose genes negatively correlated with the COMD coefficient across tissues (Gene set enrichment analysis, False Discovery Rate, FDR < 10[-6] for Mitochondrial ATP synthesis coupled electron transport and other mitochondrial ATP synthesis related GO terms, Fig. 2A,B; Dataset EV1). Furthermore, this negative correlation was also observed across individuals within the same tissues (Fig. 2C; Appendix Fig. S5C). To assess the generality of these observations beyond the GTEx dataset, we next processed single-cell RNA-Seq data from 45,146 mouse cells (Almanzar et al, 2020) that we aggregated into 45 cell types. In mouse cells, mitochondrial ATP synthesis genes also negatively correlated with the COMD coefficient (Gene set enrichment analysis, FDR < 0.01, Appendix Fig. 5A–C). Taken together, these observations suggest

that the more reduced the mitochondrial ATP synthesis, the more beneficial to mRNA stability the usage of optimal codons. Consistent with this hypothesis, we furthermore found that age, which is linked with a decline in mitochondrial function (reviewed in Sun et al (2016)), positively correlated with the COMD coefficient (Fig. 2D, GTEx dataset).

Translation is one of the most energy-demanding processes in the cell (Buttgereit and Brand, 1995). In order to decode one codon at least 3 energy-carrier molecules are needed, 1 ATP and 2 GTP (Dever et al, 2018; Gomez and Ibba, 2020). The previous observations indicate a link between codon optimality and ATP production. We then set out to test through perturbation analyses whether this link was causal or merely correlative. The GTEx dataset, consisting of post-mortem samples, provides data for which Nature performed such a perturbation experiment. At death, respiration and blood circulation cease, which deprives the body's cells of oxygen and impairs mitochondrial ATP synthesis. Moreover, the GTEx samples were stabilized at different times after death, or ischemic times, allowing us to study the effects of varying degrees of oxygen deprivation. We found the COMD coefficient to increase with ischemic time adjusting for age and tissue (multivariate analysis Fig. 3A, stratification Fig. 3B, Appendix Figs. S6–10). These results, which show that the effect of codon optimality on mRNA stability is amplified upon oxygen deprivation, agree with the role of ATP production implied by the previous observations in GTEx and mouse.

However, ischemic times in the GTEx dataset span hundreds of minutes. It cannot be excluded that mechanisms other than intracellular ATP deprivation (Ferreira et al, 2018), including tRNA regulatory response, could be occurring during these times. To address this concern, we next performed an experiment to assay the short-term response of codon optimality effects to ATP deprivation. To this end, we performed 5′P sequencing (5PSeq) (Pelechano et al, 2016; Zhang and Pelechano, 2021) on S. cerevisiae cultures following exposure to antimycin A, a cellular respiration inhibitor (Fig. 3C). 5PSeq maps the 5′ ends of RNA fragments resulting from 5′ to 3′ degradation which is generally carried out by XRN1 and occurs predominantly co-translationally (Pelechano et al, 2016). In this case, the ribosome protects the RNA from XRN1 degradation, making 5PSeq a toeprinting assay for ribosome position mapping (Fig. 3D). Variations in codon-associated 5Pseq coverage mostly come from ribosome occupancy changes (Pelechano et al, 2016; Zhang and Pelechano, 2021). We derived a measure, which we termed codon-associated 5′ coverage, that captured how much each codon associates with the abundance of 5′ RNA degradation intermediates corresponding to its A-site position while adjusting for the 17-nt shift due to ribosome protection as well as controlling for gene coverage, distance to the start codon, and sequencing depth (Fig. 3D for raw data around a non-optimal codon, Methods). As expected, these estimates of codon-associated 5′ coverage were consistent with previously reported decoding times and codon optimality in yeast (Appendix Fig. S11). Notably, ratios in codon-associated 5′ coverage between optimal and non-optimal codons negatively correlated with intracellular ATP concentration over the time course (Fig. 3E,F). These results are consistent with ATP concentration modulating the contribution of codon optimality to translation and translation-related mRNA degradation. Furthermore, these observations made within a few minutes upon ATP deprivation, cannot be explained by a transcriptional response of tRNA abundance.

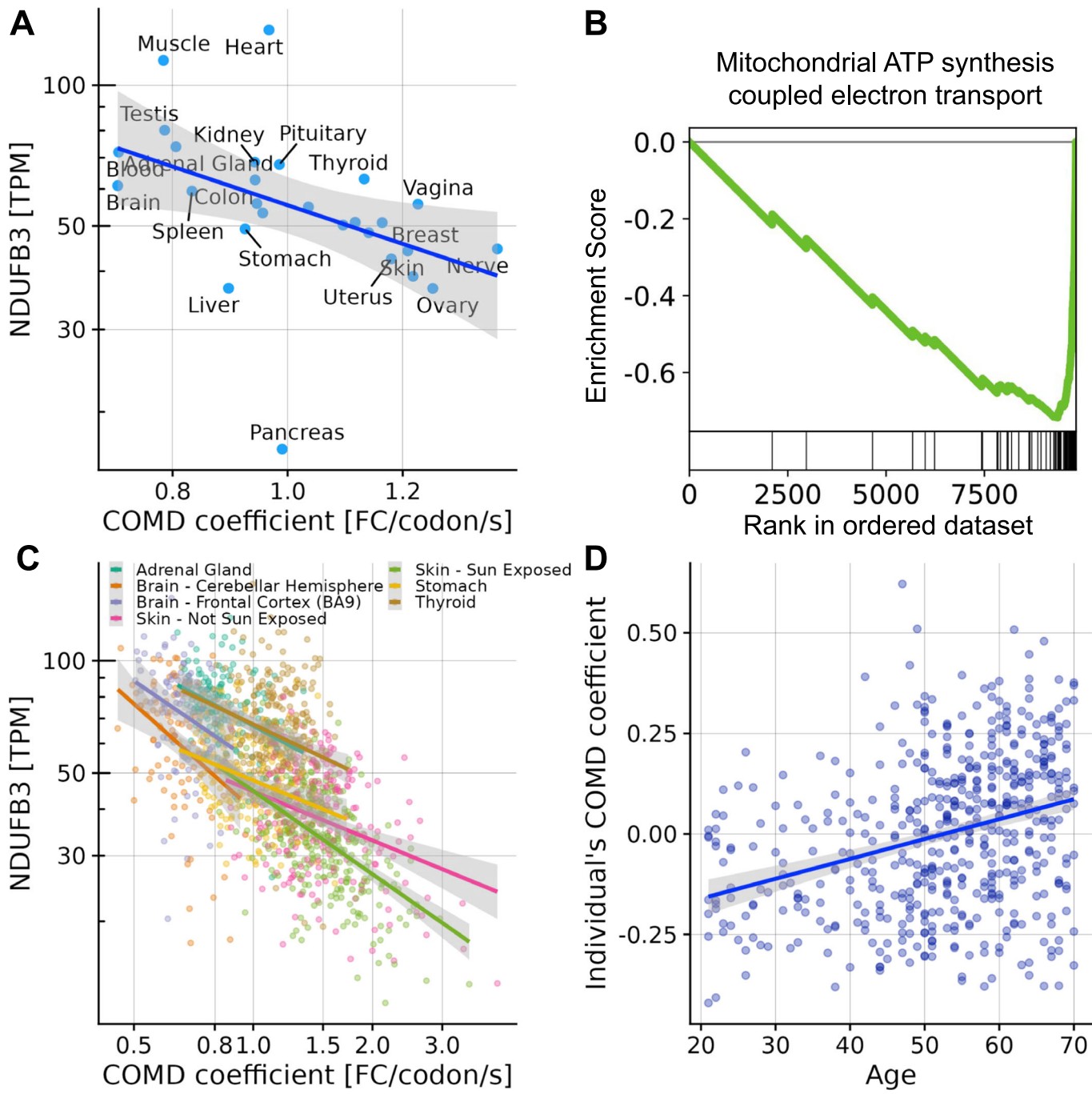

**Figure 2. Mitochondrial ATP synthesis activity associates with the COMD coefficient between tissues and individuals.**

(A) Expression in transcripts-per-million (TPM) across major GTEx tissues of *NDUFB3*, a representative nuclear-encoded gene that encodes a protein of the mitochondrial respiratory chain, against the COMD coefficient (Spearman's rho $= -0.62$, $P = 7.5 \times 10^{-4}$). (B) Gene set enrichment analysis (Subramanian et al, 2005) for the Gene Ontology biological process "mitochondrial ATP synthesis coupled electron transport" in human. (C) *NDUFB3* expression (TPM) against the COMD coefficient across GTEx individuals for 7 tissues. (D) COMD coefficient of GTEx individuals estimated adjusting for tissue (Methods) against age (Spearman's rho $= 0.33$, $P = 9 \times 10^{-15}$).

Our findings reveal a fundamental link between metabolism, gene expression, and gene sequence, the consequences of which need to be explored. As a first step in this direction, we asked whether this phenomenon could constrain tissue-specific mRNA isoform regulation. To this end, we considered cassette exons, i.e. exons located in between two other exons, exhibiting a tissue-specific inclusion pattern. In sun-exposed skin, a tissue with a high COMD coefficient and low mitochondrial activity, we found that cassette exons that were more often included used more optimal codons compared to cassette exons that were more often skipped (Fig. 4A). The opposite was observed in the cerebellar hemisphere, a tissue with a low COMD coefficient and high mitochondrial

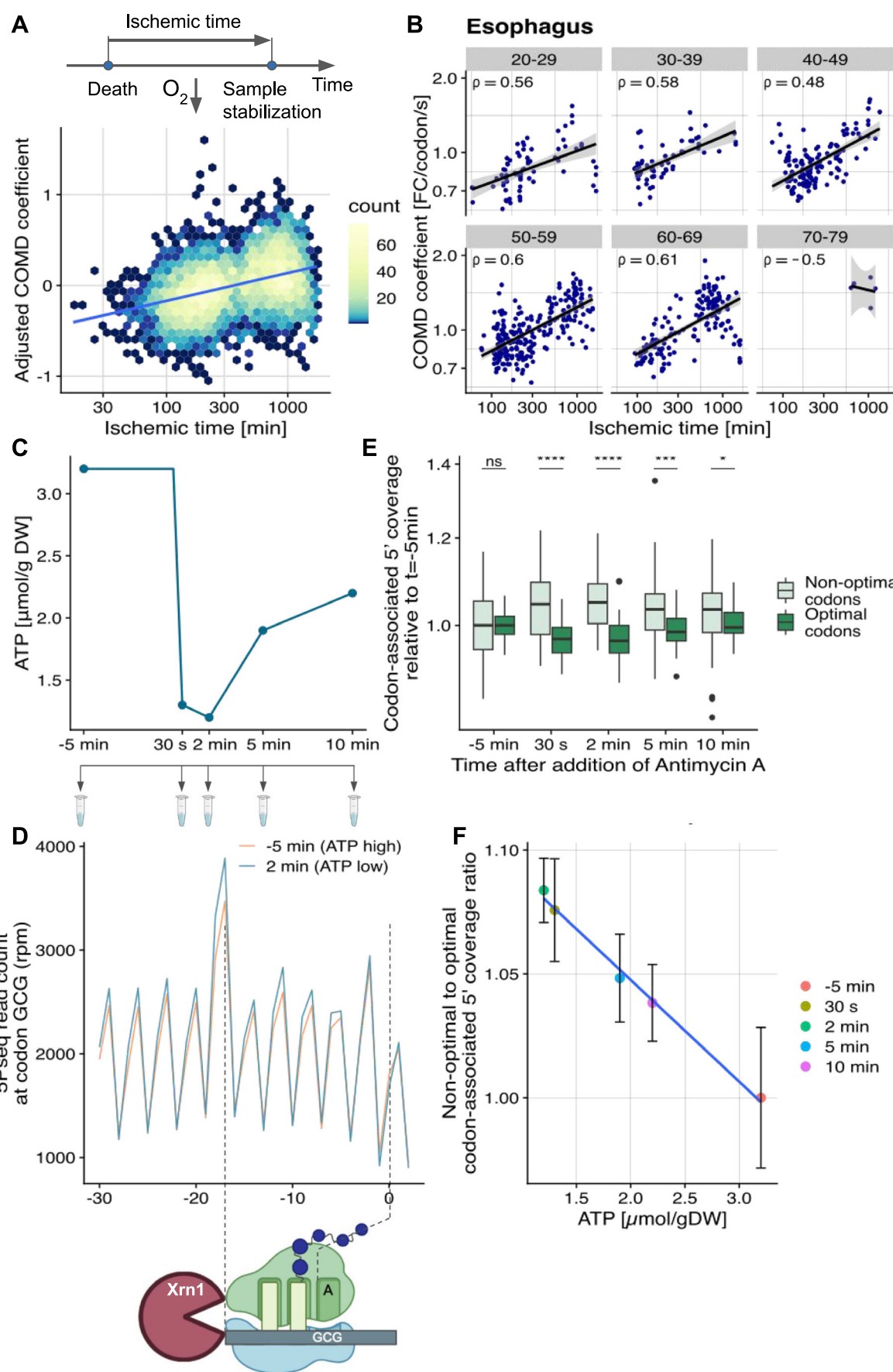

**Figure 3.   Differences in the contribution to mRNA degradation of optimal and non-optimal codons depend on intracellular ATP concentration.**

(**A**) COMD coefficient adjusted for age and tissue (Methods) against ischemic time (Spearman's rho = 0.41 $P < 10^{-15}$). Blue line marks linear trend obtained by linear regression. (**B**) COMD coefficient against ischemic time for esophagus samples grouped by age category (Spearman's rank correlation rho is statistically significant ($P < 0.05$) for all age categories but 70–79). (**C**) Sampling design of 5PSeq profiling time course following addition of the cellular respiration inhibitor Antimycin A on yeast cells. For each time point three biological replicates were generated. The corresponding intracellular ATP concentration was taken from Walther et al, 2010. (**D**) Number of 5P-seq reads (in reads per million) per position relative to the GCG codons for the time points with maximum (−5 min) and minimum (2 min) intracellular ATP concentration. 5PSeq predominantly maps the 5′ends of mRNAs co-translationally degraded by Xrn1, which are located 17 nucleotides 5′ of the ribosome A site (Pelechano et al, 2015). Hence, the peak located 17 nt 5′ of GCG codons is consistent with GCG being a non-optimal codon in yeast. This peak is amplified at low ATP concentration (2 min time point). (**E**) Distribution of codon-associated 5′ coverage inferred from 5PSeq for non-optimal and optimal codons (defined using first and fourth quartiles of occupancy in unperturbed cells). Values are expressed in fold-change relative to median codon effect in unperturbed cells (Methods). *P*-values were obtained from double-sided Wilcoxon rank-sum tests (*<0.05, **<0.01, ***<0.001, ****<0.0001). For each boxplot: Center line, median; box limits, first and third quartiles; whiskers span all data within 1.5 interquartile ranges of the lower and upper quartiles. The number of points for each boxplot is $n = 45$. (**F**) Ratio of codon-associated 5′ coverage for each time point between non-optimal and optimal codons with error bars representing ± 1 standard deviation based on a permutation test ($n = 1000$ points from 1000 permutations), alongside its corresponding reported intracellular ATP concentration (Walther, 2010).

activity. Furthermore, the trend generalized to tissues with intermediate COMD coefficient values (Fig. 4B). Hence, these observations reveal an unanticipated codon usage bias for tissue-specific exons. Moreover, our observations and experiments provide an explanation for this: non-optimal codons are relatively less likely to trigger mRNA degradation in tissues with high cellular energy. Therefore, splice isoforms using non-optimal codons are comparatively more stable in tissues with high cellular energy, which is reflected in an increased abundance of cassette exons using slow codons in the mRNA pool. Perhaps, even, these exons have evolved to use codons suited to the metabolic state of the tissue they are expressed in.

How could cellular energy mechanistically modulate how much codon optimality impacts mRNA stability? Nearly all the energy required by translation is consumed during elongation (Liu and Proud, 2016) through ATP or GTP where it is used to power the codon decoding steps and tRNA recharging. Mitochondria produce energy in the form of ATP, which regenerates GTP (Boissan et al, 2018). We hypothesized that changes in ATP abundance, and therefore GTP as well, alter the kinetics of the translation elongation cycle unequally for different codons and consequently how likely the Ccr4-Not complex gets recruited and triggers mRNA degradation. To formally explore this hypothesis, we developed a mathematical model of the translation elongation cycle (Fig. 5A, Methods).

In a translation elongation cycle, the ternary complex (TC), which is composed of one aminoacyl-tRNA, one GTP, and one eukaryotic translation elongation factor 1A (eEF1A), is first loaded into the A-site of a translating ribosome. Next, the amino acid is added to the nascent polypeptide chain and the ribosome translocates, freeing up the A-site for a new cycle (Dever et al, 2018). If TC loading is slow, the E-site can be freed while the A-site remains empty, setting the ribosome into a conformation recognized by the Ccr4-Not complex (Buschauer et al, 2020). Hence, the faster the TC gets loaded on the ribosome, the less likely mRNA degradation is triggered.

Our mathematical model predicts that higher amounts of ATP increase the fraction of tRNAs in TC and, consequently, the TC loading rate (Appendix Fig. S12). Moreover, the model describes how this relationship depends on the overall abundance of the tRNA. Non-optimal codons, for which the abundance of cognate tRNAs is low (Bae and Coller, 2022), show a relatively higher TC loading rate increase as ATP increases compared to optimal

codons. As a result, the TC loading rate ratio of optimal to non-optimal codons decreases with ATP concentration (Fig. 5B). This qualitative behavior was robust to choices of rate constants within a broad range of plausible values (Fig. 5C). Altogether, these theoretical investigations support a model in which cellular energy attenuates the effects of codon optimality by dampening the impact of overall cognate tRNA abundance variation on TC loading.

## Discussion

Taken together, our results show that cellular energy regulates the effect of codon optimality on mRNA stability. Codon optimality matters more in conditions of scarcer energy, such as tissues with low mitochondrial activity, older age, oxygen deprivation, or exposure to specific drugs. In addition, we found this effect to be reflected in the codon usage of tissue-specific cassette exons.

Previous studies reported differences in the effect of codons on gene expression for cells in distinct proliferation states (Guimaraes et al, 2020; Gingold et al, 2014). However, regulation of the tRNA pool as a driver of such differences, although observed, is not consensual. Our study provides an explanation for the reported attenuated effects of codon usage in proliferative cells in absence of tRNA regulation (Guimaraes et al, 2020), as ATP concentration is maximal at the G2/M phase (Marcussen and Larsen, 1996).

The experimental validations were conducted using yeast as a model system, which may not properly capture the relevant biology of human cells, posing a limitation to our study. Nevertheless, the generality of our findings for diverse eukaryotes is supported by the consistent association of the effects of codon optimality with mitochondrial gene expression in human and mouse, as well as the attenuated effects of codon optimality in brain and testis previously reported for the fruit fly (Burow et al, 2018b; Allen et al, 2022)—an observation for which we now provide an explanation.

A second limitation to our study is that we indirectly used the expression of genes involved in mitochondrial ATP production as a proxy for the cellular energy status when analyzing the human and mouse transcriptome data. ATP concentration for different tissues in humans are scarce and the reported values are highly variable (Greiner and Glonek, 2021). Using yeast as a model organism allowed us to probe a simpler organism where the ATP abundance response to perturbations in ATP production has been well

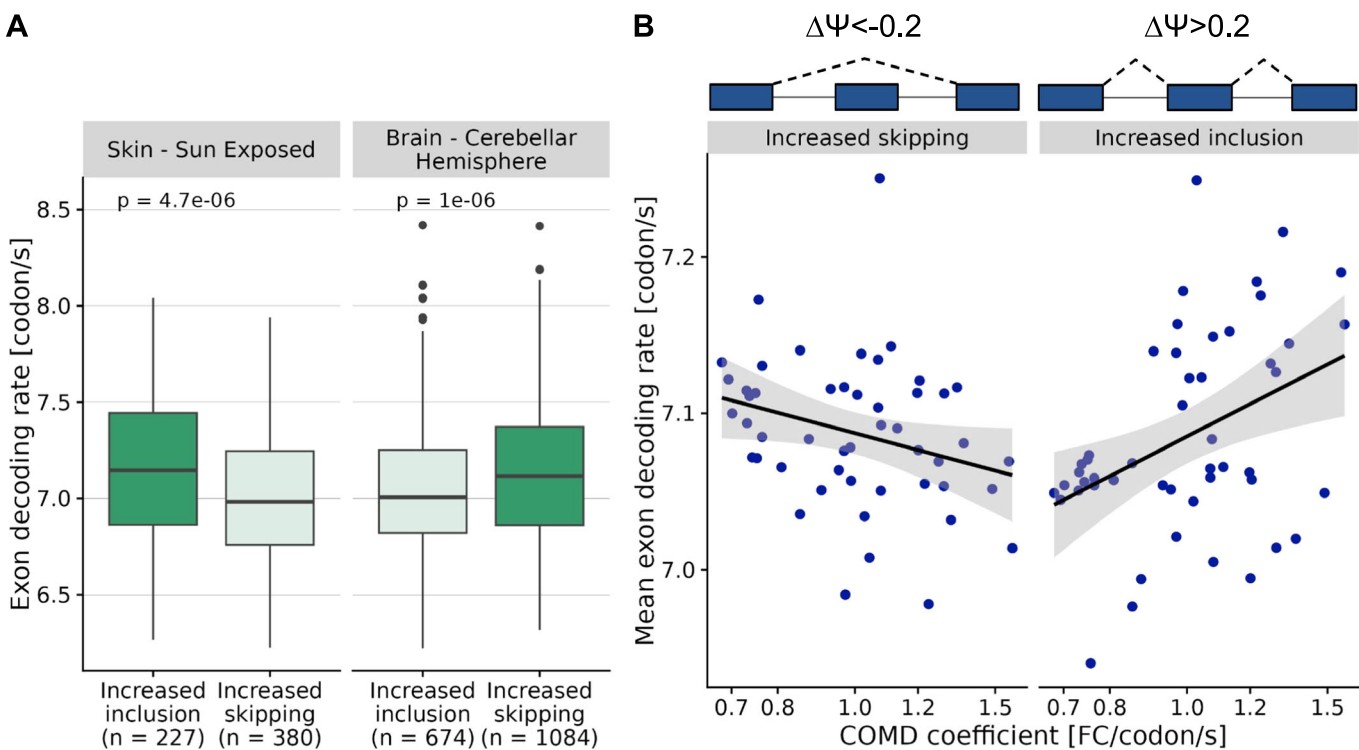

**Figure 4.  Codon optimality of cassette exons increasingly included or skipped in a tissue relates to its COMD coefficient.**

(A) Distribution of the decoding rate of cassette exons that are increasingly included ($\Delta\Psi > 0.2$) or skipped ($\Delta\Psi < -0.2$) in sun-exposed skin (left) and cerebellar hemisphere (right) compared to other tissues. Exon decoding rates are expressed in codons per second and computed using codon decoding rates measured in HEK293 cells (Methods). *P*-values were obtained from a Wilcoxon rank-sum test. For each boxplot: Center line, median; box limits, first and third quartiles; whiskers span all data within 1.5 interquartile ranges of the lower and upper quartiles. The number of points for each boxplot is indicated in the figure labels. (B) Codon decoding rate average of increasingly skipped (left) and included (right) exons per tissue against the tissue's COMD coefficient. One-sided Spearman's rank correlation rho is statistically significant for both panels ($P = 0.018$ and $P = 0.007$, one-sided *P*-value).

characterized. Moreover, using yeast made it possible to perform 5PSeq, a method to toeprint the ribosome in vivo without the need for translation inhibitors. An alternative to 5PSeq for profiling ribosome occupancies could have been ribosome sequencing (Ribo-Seq). However, Ribo-Seq requires larger sample volumes, making the study of rapid phenomena occurring difficult. In contrast, 5PSeq can be executed with just 1 mL of sample volume, enabling the utilization of rapid, Eppendorf-based, collection methods. Moreover, 5PSeq leverages the intrinsic toeprinting activity of endogenous RNases active in the cell and does not require polysome fractionation, in vitro RNA digestion, or other extensive manipulations. Therefore, 5PSeq can be more easily performed at very short times by using flash freezing followed by sequencing. Hence, 5PSeq allowed us to profile reactions within minutes, before widespread transcriptional response to the toxicity of ATP production inhibition and before transcriptional response of tRNA synthesis. One limitation of 5PSeq is that not all reads originate from ribosome toeprinting. The coverage of 5PSeq along the coding sequence also entails from RNA that are not co-translationally degraded. Hence, increased translation-independent degradation could make 5PSeq coverage less dependent on ribosome occupancy and consequently on codon identity. Under this alternative model, the attenuated effect of codon optimality for high ATP observed in our 5PSeq experiment would be due to an increase in the strength of translation-independent RNA degradation over COMD. Future

studies are needed to distinguish between a mechanism for which ATP modulates the relative strength of COMD versus translation-independent degradation on the one hand, and ATP-dependent codon decoding rate as we mathematically modeled here on the other hand.

A third limitation of our study is that we did not assay mRNA stability variations directly. For the mammalian transcriptomes we used fold-change of the exonic-to-intronic read count ratio as a proxy. Tissue-specific increased persistence of intronic reads could affect exonic-to-intronic read count ratio. Importantly, the COMD coefficient remains unaffected assuming such factors are not gene-specific and act multiplicatively. More complex non-multiplicative relationships could exist, although it is unclear how they could take shape. Also, we did not measure RNA stability changes in the yeast experiment. On the one hand, measuring mRNA stability under such dynamic and short time changes is technically challenging (Uvarovskii et al, 2019). On the other hand, even if it were technically feasible, ribosome occupancy changes may not lead to measurable mRNA stability changes under this short time period.

Our theoretical investigations suggest TC loading as one possible biochemical pathway linking cellular energy to codon optimality. In this mathematical model, the abundance of energy carrier molecules differentially modulates the proportion of tRNAs in a ternary complex which affects codon decoding. Ultimately, this would allow codon

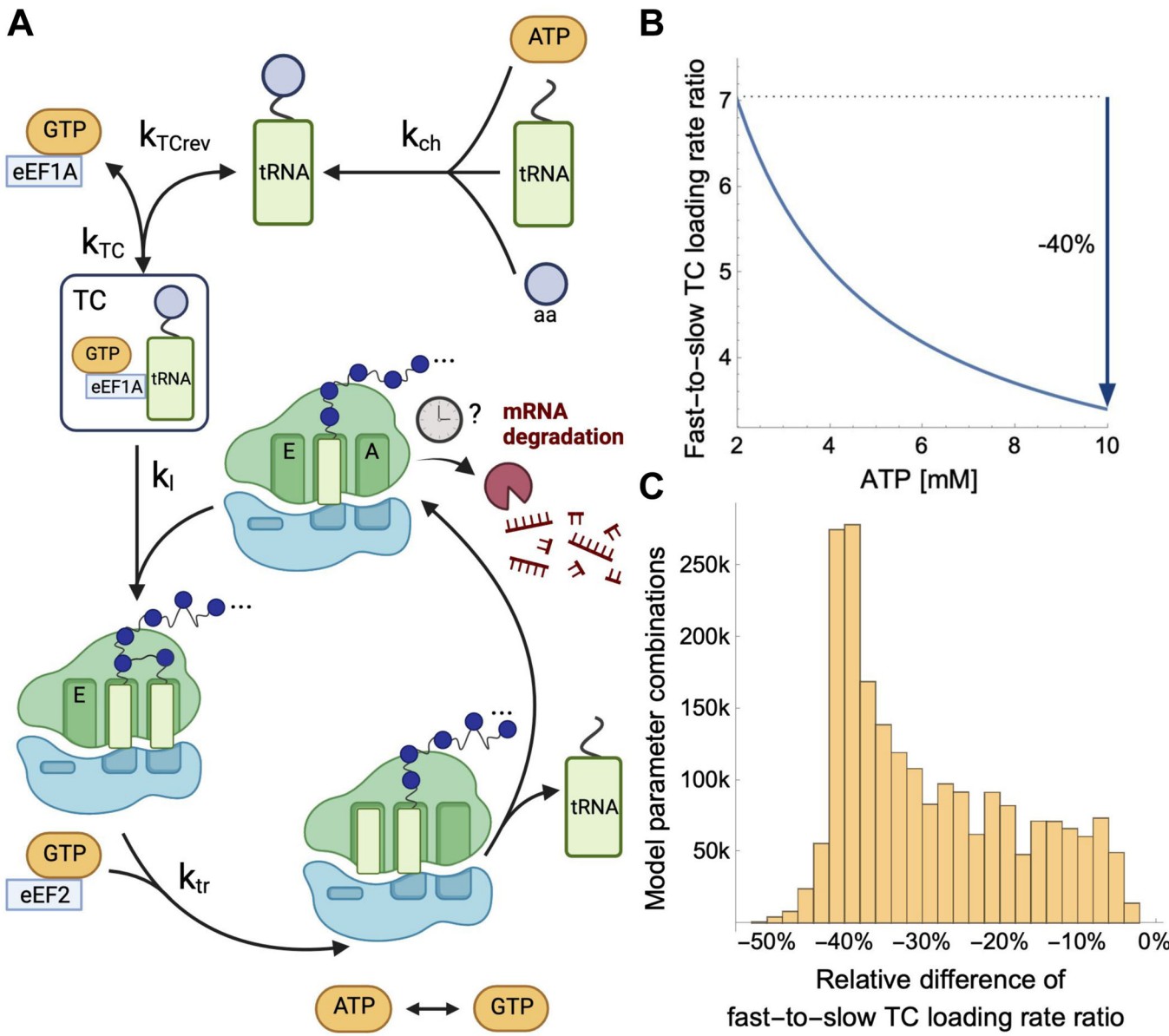

**Figure 5. Mathematical modeling predicts that ATP concentration modulates differences in decoding rates between codons.**

(A) Reactions considered in our mathematical model of the eukaryotic translation elongation cycle (top to bottom): tRNA aminoacylation, ternary complex (TC) formation, TC loading, ribosome translocation, and ATP to GTP conversion. The mRNA degradation is depicted but not included in the model. (B) Fast-to-slow TC loading rate ratio against ATP concentration for a representative combination of plausible kinetic rate constants (Methods). The mathematical model predicts that the fast-to-slow TC loading rate ratio is 40% lower for the maximal compared to the minimal ATP concentration. (C) Distribution of the relative difference between the minimal and maximal ATP concentrations of the fast-to-slow TC loading rate ratio across combinations of plausible model parameter values. Across all parameter combinations, the TC loading rate ratio is lower for maximal ATP concentrations.

decoding rates to be modulated despite reported stable proportions of the tRNA pool per anticodon across tissues (Schmitt et al, 2014; Pinkard et al, 2020; Behrens et al, 2021). Indeed, our mathematical model shows that there is a gap between the total amount of tRNAs and the proportion of tRNAs in a ternary complex and therefore ready to be loaded in the ribosome. This is consistent with a previous study in *E. coli* that reports that the proportion of tRNAs in a ternary complex is highly variable and can range from 8% to 87% depending on the tRNA (Rudorf and Lipowsky, 2015).

Importantly, our mathematical model does not exclude alternative mechanisms. As mentioned above, one possibility is that higher cellular energy levels could increase the rate of translation-independent mRNA degradation which would attenuate differences arising from codon identity.

Another possibility would be that higher cellular energy levels increase the sensitivity of ribosome pausing to degradation, making the triggering of mRNA degradation less dependent on codon identity. However, there is no report showing that monitoring of codon

optimality is energy-dependent. Although to a lesser extent than for translation elongation, ATP is also required in translation initiation (Jackson et al, 2010), termination and ribosome recycling (Dever and Green 2012), and by helicases that unwind RNA structures facilitating translation initiation and potentially other translation steps (Parsyan et al, 2011). One cannot exclude that effects of ATP variation on those steps may be reflected in transcriptome-wide variation of codon-specific ribosome occupancies. Moreover, dedicated pathways sensing ATP levels involved in controlling translation may shape the effect of codon optimality. This could be the case for the mTORC1 pathway, which controls translation given diverse cellular cues such as energy status, oxygen availability, and amino-acid levels (Laplante and Sabatini, 2012; Darnell et al, 2018).

In the broader context, our findings relate to other studies linking cell physiology and homeostasis with mRNA degradation. In particular, mRNA degradation has been found to regulate mRNA abundance either globally, to buffer changes in mRNA abundance, or specifically to regulate sets of genes (Sun et al, 2012; Swaffer et al, 2023; Eser et al, 2014; García-Martínez et al, 2021). Whether and how our observations relate to these phenomena remains to be investigated. For instance, the buffering of mRNA levels seen upon transcriptional changes or in cells with higher volumes could in part come from attenuated codon optimality resulting in lower mRNA degradation under higher cellular energy. Moreover, the potential biological mechanisms linking cellular energy to codon optimality could take place to varying degrees and are not mutually exclusive. Identifying which mechanisms take place and delineating their contribution requires further research.

Altogether, our work uncovers a fundamental link between cellular energy and eukaryotic gene expression, alternative or complementary to tRNA pool regulation. The functional implications of this link extend beyond tissue and age specific to altered energy metabolism states, such as in cancer and specific cellular environments, potentially providing a novel way for these abnormal states to shape gene expression.

# Methods

### GTEx exonic and intronic read counts

The BAM files for 7778 RNA-Seq samples, the gene-level and the transcript-level TPM (transcript per million) values, as well as the sample annotation of the GTEx v6 dataset, genome build hg19, were downloaded from the GTEx portal on June 12, 2017, under accession number dbGaP: phs00424.v6.p1. This RNA-Seq dataset is paired-end and unstranded. Exonic coordinates of all protein-coding genes located in standard chromosomes were extracted from the GENCODE annotation (Frankish et al, 2021), release 19. Exonic and intronic read counts were obtained as recommended by (Gaidatzis et al, 2015). Specifically, exonic coordinates were flanked on both sides by 10 nt and were grouped by gene. Intronic coordinates were obtained by subtracting the exonic coordinates from the gene-wise coordinates. For each gene, exonic and intronic read counts were obtained using the summarizeOverlaps function from the GenomicAlignments package (Lawrence et al, 2013) v. 1.28.0 with the mode parameter set to "IntersectionStrict" and the inter.feature parameter set to FALSE to consider only reads that fully fall within the desired genomic regions. Moreover, to be robust against noisy estimates based on

low read counts, in each sample genes with TPM < 1 were ignored (read counts set as missing values). Finally, for each gene and each sample, the log-transformed exonic-to-intronic read count ratio $y$ was computed using pseudocounts of 1:

$$y := \log_2(\text{exonic counts} + 1) - \log_2(\text{intronic counts} + 1).$$

### Relative mRNA half-life

Next, we attributed those $y$ values to the major isoform for each sample, whereby the major transcript isoform was taken as the transcript isoform with highest median TPM value across samples of the same tissue. Transcripts with missing values in more than one third of samples were discarded. For the remaining transcripts, tissue-specific log-transformed exonic-to-intronic read count ratio were calculated by taking the median of the $y$ values over all samples from the same tissue. We further filtered out transcripts with missing values in more than 15% of the tissues.

Exonic and intronic read counts depend on gene-specific biases including exonic and intronic length and GC-content. Following (Gaidatzis et al, 2015), these biases are mostly multiplicative and therefore cancel out when considering ratios between samples. We therefore use $y$ differences as estimates for mRNA half-life log-ratios between pairs of samples or tissues. Following the same logic, the $\log_2$ relative mRNA half-life $\widetilde{y_{ij}}$ for transcript $i$ in sample or tissue $j$ was defined relatively to the average across tissues or samples, respectively, i.e.:

$$\widetilde{y_{i,j}} := y_{i,j} - \overline{y_i}$$

The entire procedure was applied at two levels of tissue annotation granularity: the GTEx major tissues on the one hand, and the GTEx subtissues on the other hand.

### Transcript sequence features

Transcript sequences were retrieved using pyranges v0.0.84, kipoiseq v0.4.1 from the release 19 of the GENCODE annotation of the human genomic sequence GRCh37/hg19. Only coding sequences starting with the start codon AUG were considered.

### Codon effects

The estimated codon effect, $s_{k,j}$, was obtained for each codon $k$ and tissue $j$ separately by fitting a univariate linear regression of the type:

$$\widetilde{y_{i,j}} = s_{k,j} f_{k,i} + \varepsilon_{i,j,k},$$

where $f_{k,i}$, is the proportion of codon $k$ in the coding sequence of transcript $i$. To this end, we used scikit-learn v0.22.2.

### COMD coefficient

The COMD coefficient was obtained for each tissue or sample $j$ by fitting a linear regression of the type:

$$\widetilde{y_{i,j}} = \alpha_j r_i + \varepsilon_{i,j},$$

where $\alpha_j$ is the COMD coefficient for the sample or the tissue $j$ and $r_i$ is the geometric mean of the codon decoding rates of transcript $i$ in HEK293.

## Generation of spliced and unspliced read counts ratio for the mouse dataset

The BAM files and valid barcodes for the 28 scRNA-Seq samples (using droplet-based 10x Genomics Chromium protocol) of 3-month-old mice of the Tabula Muris Senis atlas dataset (Almanzar et al, 2020) were downloaded from the public AWS S3 bucket (https://registry.opendata.aws/tabula-muris-senis/). The reference genome used to generate the BAM files did not contain mitochondrial-encoded genes, therefore they were not considered for our analysis. Sample annotations were downloaded from the Gene Expression Omnibus under accession code GSM4505404, specifically the file GSM4505404_tabula-muris-senis-droplet-official-raw-obj.h5ad.

Loom files for each sample containing raw spliced and unspliced counts were obtained by running the velocyto command-line tool ((La Manno et al, 2018), v0.17.17). In contrast to bulk RNA-Seq, UMI-based scRNA-Seq (unique molecular identifier) allows identifying whether reads originated from the same transcript. The velocyto tool makes use of this and collectively marks reads of the same transcript as unspliced if one of them aligns to an intronic region or exon-intron junction. Conversely, if all reads of the same UMI solely align to exonic regions, they are marked as spliced reads. Equivalently to exonic and intronic reads, unspliced reads are a proxy for premature mRNA and spliced reads a proxy for mature mRNA, therefore their ratio is a proxy for mRNA stability (Gaidatzis et al, 2015).

In order to have enough cells for pseudo-bulking, we filtered out cell types ("cell_ontology_class") that had less than 150 cells using scanpy (Wolf et al, 2018), v1.8.2). For the remaining 45 cell types, we computed pseudo-bulk aggregates by summing all counts of cells per cell type for spliced and unspliced counts independently. To only consider genes that are expressed across multiple cell types, we filtered out genes with less than 3000 counts shared across spliced and unspliced counts and all cell types. We normalized both spliced and unspliced counts by dividing them by the total number of spliced and unspliced counts, respectively, over all genes per cell type. Per gene spliced-to-unspliced ratios were computed as $\log_{10}(\text{spliced counts} + 1) - \log_{10}(\text{unspliced counts} + 1)$. Spliced-to-unspliced ratios were centered across cell types for each gene. As gene expression, we used the normalized and log-transformed spliced counts.

## Transcript sequence features for the mouse dataset

Transcript features for mouse were retrieved as described for human. For mouse, we used release 25 of the GENCODE annotation (Frankish et al, 2021) of the mouse genomic sequence GRCm38/hg38.

## Major transcript isoform selection for the mouse dataset

Since the 10x single-cell technology does not produce full-length transcript coverage, the major transcript of a gene was defined as the transcript with the highest support. We considered the tags of each transcript (GENCODE, release 25) and calculated the support

by summing up the following categories: being part of the GENCODE basic annotation (tag "basic"), being tagged as the principal isoform according to APPRIS database (tag "appris_principal_1", (Rodriguez et al, 2013)), being a member of the consensus CDS gene set (tag "CCDS", (Pujar et al, 2018)), and having an Ensembl transcript support level of 1 ("transcript_support_level"). A transcript could therefore have a support between 0 and 4 and for each gene we chose the transcript with the highest support. If there were ties, one transcript was randomly chosen.

## Gene set enrichment analysis

Gene set enrichment analysis (Subramanian et al, 2005) was performed on the genes scored by their Spearman correlation between their TPM values and the COMD coefficient across tissues using GSEApreranked with package gseapy v1.0.0. Only genes with TPM > 1 in all tissues were considered. Within-tissue correlation between the COMD coefficient and gene expression (TPM) followed by GSEA was computed as before. Only genes with TPM > 1 in at least two-thirds of the samples were considered.

In mouse, we considered genes whose log-transformed and normalized spliced counts were greater than 0 across all cell types (6051 genes). As in human, gene set enrichment analysis was performed on the genes scored by their Spearman correlation.

For all GSEA analyses, only pathways with FDR $\leq$ 0.01 were considered significantly enriched.

## Individual's COMD coefficient

We fitted a linear regression predicting the COMD coefficient of each sample as the sum of an individual's effect (which we defined as the individual's COMD coefficient) and a tissue effect.

## Overview of the elongation cycle model

We developed a simplified steady-state model of one translation elongation cycle to assess how variations in ATP and GTP abundance can change the rate of ternary complex loading into the ribosome for given tRNA and ribosome concentrations. We modeled the 2-1-2 pathway of E-site tRNA release, because, unlike the 2-3-2 pathway, it includes the state of the ribosome in which both the A-site and the E-site are free (Chen et al, 2011), which is the substrate for the Ccr4-Not complex (Buschauer et al, 2020).

In our simplified model, we considered the ribosome in two states: free A-site or occupied A-site. The free A-site state corresponds to the point where the ribosome is ready to accept its cognate ternary complex. In this state the ribosome has finished translocation, the P-site is occupied and both the A-site and the E-site are free.

The occupied A-site state represents the point where the ribosome is ready to translocate. In this state, the tRNA has already been accommodated into the A-site (following GTP hydrolysis) and the new peptide bond formed. In this state, both the A-site and the P-site are occupied and the E-site is free.

The transition between the free and the occupied A-site states described above is characterized by a series of intermediate states, such as peptide-bond formation and ribosome conformation changes (reviewed by Dever et al (2018)), that do not depend on

the variables of interest: tRNA, ribosome, GTP and ATP concentrations. Therefore, under changes in these variables, the rate of transition between such intermediate states gives a constant contribution to the rate of transition between the free and occupied A-site states. Following from this, we considered the following reactions: tRNA aminoacylation, ternary complex formation, ternary complex loading, ribosome translocation (Fig. 2E).

## tRNA aminoacylation

tRNAs are charged with amino acids by the aminoacyl-tRNA-synthetase in a two-step reaction, where ATP is hydrolyzed to AMP and one amino acid is loaded into the tRNA (reviewed by Gomez and Ibba (2020)). We modeled tRNA aminoacylation with an overall irreversible reaction:

$$\text{tRNA}_{\text{free}} + \text{ATP} + \text{aa} \rightarrow \text{aa-tRNA} + \text{AMP} + \text{PPi}$$

where $\text{tRNA}_{\text{free}}$ represents the pool of uncharged tRNAs available to be aminoacylated.

Assuming law of mass action, we modeled the rate of tRNA aminoacylation as:

$$v_{\text{tRNA aminoacylation}} = k_{\text{ch}}[\text{ATP}][\text{tRNA}_{\text{free}}], \tag{1}$$

where $k_{\text{ch}}$ is the rate constant of the tRNA aminoacylation reaction under a given concentration of amino acids.

## Ternary complex loading

The ternary complex, composed of one aminoacyl-tRNA, one GTP, and one eukaryotic translation elongation factor 1A (eEF1A) binds in the A-site of the ribosome, where the aminoacyl tRNA is accommodated after the hydrolysis of GTP followed by the release of eEF1A-GDP (Dever et al, 2018). We modeled the TC binding to the A-site of the ribosome and the subsequent tRNA accommodation as a single irreversible reaction which we termed TC loading:

$$\text{TC} + \text{Ribosome}_{\text{free A-site}} \rightarrow \text{Ribosome}_{\text{occupied A-site}} + \text{eEF1A-GDP} + \text{Pi}$$

We assumed law of mass action, i.e:

$$v_{\text{TC loading}} = k_l[\text{TC}][\text{Ribosome}_{\text{free A-site}}], \tag{2}$$

where $k_l$ is the rate constant of the TC loading reaction.

## Ribosome translocation

Translocation of the ribosome requires the binding of eEF2-GTP to the A-site of the ribosome which is followed by the hydrolysis of GTP and subsequent release of eEF2-GDP. After translocation, the deacylated tRNA on the E-site is released from the ribosome (Dever et al, 2018). We modeled this process as a single irreversible reaction combining the recharging of eEF2 with GTP and its subsequent binding to the ribosome A-site followed by GTP

hydrolysis, which results in ribosome translocation. Furthermore, we combined together ribosome translocation and the release of tRNA from the E-site:

$$\text{Ribosome}_{\text{occupied A-site}} + \text{eEF2} + \text{GTP} \rightarrow \text{Ribosome}_{\text{free A-site}} + \text{tRNA}_{\text{free}} + \text{eEF2-GDP} + \text{Pi}$$

We assumed law of mass action kinetics, i.e.:

$$v_{\text{translocation}} = k_{\text{tr}}[\text{GTP}]\left[\text{Ribosome}_{\text{occupied A-site}}\right], \tag{3}$$

where $k_{\text{tr}}$ is the rate constant of the translocation reaction under a constant concentration of eEF2.

## Ternary complex formation

Aminoacyl-tRNAs (aa-tRNAs) are bound to the ribosome in a ternary complex (TC) with GTP and eEF1A (elongation factor 1A). eEF1A is charged with GTP by the exchange factor eEF1B (Dever et al, 2018). We modeled the eEF1A charging with GTP and subsequent binding to aa-tRNA into a single reversible reaction:

$$\text{aa-tRNA} + \text{eEF1A} + \text{GTP} \rightleftharpoons \text{TC}$$

We assumed law of mass action kinetics, i.e.:

$$v_{\text{TC formation}} = k_{\text{TC}}[\text{GTP}][\text{aa-tRNA}] \tag{4}$$

and

$$v_{\text{TC dissociation}} = k_{\text{TC}_{\text{rev}}}[\text{TC}], \tag{5}$$

where $k_{\text{TC}}$ and correspond to the rate constant of the TC formation and TC dissociation reactions, respectively, under a constant concentration of eEF1A.

## Steady-state solution

Assuming steady state, we symbolically solved the resulting system of equations in Wolfram Mathematica v13.0.0.0. This led to the following expression for the rate of TC loading as a function of ATP, GTP, $\text{tRNA}_{\text{total}}$, $R_{\text{total}}$ and all rate constants considered above:

$$v_{\text{TC loading}} = \frac{1}{A}\left(B + C + \sqrt{D + (B + C)^2}\right), \tag{6}$$

where:

$$A = 2[\text{ATP}]k_{\text{ch}}k_l\left(k_{\text{TC}} + \frac{k_{\text{TCrev}}}{[\text{GTP}]}\right)$$

$$B = -[\text{GTP}]k_{\text{TC}}k_{\text{tr}}([\text{ATP}]k_{\text{ch}} + k_l[\text{R}_{\text{Total}}])$$

$$C = -[\text{ATP}]k_{\text{ch}}(k_{\text{TCrev}}k_{\text{tr}} + k_{\text{l}}(k_{\text{TC}} + k_{\text{tr}})[\text{R}_{\text{Total}}] - k_{\text{l}}k_{\text{TC}}[\text{tRNA}_{\text{Total}}])$$

$$D = 2k_{\text{ch}}k_{\text{TC}}k_{\text{tr}}[\text{ATP}][\text{GTP}][\text{tRNA}_{\text{Total}}]A$$

the total number of ribosomes is:

$$\text{R}_{\text{Total}} = \text{Ribosome}_{\text{occupied A-site}} + \text{Ribosome}_{\text{free A-site}}, \tag{7}$$

and the total number of tRNAs is:

$$\text{tRNA}_{\text{Total}} = \text{tRNA}_{\text{free}} + \text{aa-tRNA} + \text{TC} + \text{Ribosome}_{\text{occupied A-site}}. \tag{8}$$

The order of magnitude of each rate constant was estimated based on previous reports and mostly taken from *S. cerevisiae* when possible. To ensure our results are not qualitatively sensitive to the choice of rate constant, we considered multiple values spanning 2–3 orders of magnitude around the ones based on literature.

In *S. cerevisiae*, the rate of tRNA aminoacylation is in the order of magnitude of 1 tRNA per second (Chu et al, 2011) and ATP concentration can vary between 2 and 6 mM (Takaine et al, 2022). Following Eq. 1, we can infer that $k_{\text{ch}} \approx 10^3 \, \text{s}^{-1} \, \text{M}^{-1}$.

From (Trösemeier et al, 2019) the rate constant of ternary complex loading into the ribosome in *S. cerevisiae* is $k_{\text{l}} \approx 10^7 \, \text{s}^{-1} \, \text{M}^{-1}$.

The rate of translocation for one ribosome is $v_{\text{translocation}} \approx 10^2 \, \text{s}^{-1}$ (Trösemeier et al, 2019). Given the concentration of $\text{GTP} \approx 0.1$ mM (Koç et al, 2004), BNID 101420 (Milo et al, 2010) and considering the previously derived expression for the translocation rate velocity, $k_t \approx 10^6 \, \text{s}^{-1} \, \text{M}^{-1}$.

The rate of ternary complex association for one aminoacyl-tRNA was estimated to be $v_{\text{TC}} \approx 10^1 \, \text{s}^{-1}$ in *E. coli* (Rudorf and Lipowsky, 2015) assuming a constant concentration of the required elongation factor (EF-Tu) in its normal range. Given the concentration of $\text{GTP} \approx 0.1$ mM and the previously derived expression for the rate of TC formation, $v_{\text{TC}}$, $k_{\text{TC}} \approx 10^5 \, \text{s}^{-1} \, \text{M}^{-1}$.

The concentration of tRNAs in yeast commonly varies between 0.1 μM and 1 μM (Trösemeier et al, 2019). The total concentration of ribosomes is in the range 1–10 μM (Trösemeier et al, 2019). Assuming that between 1% and 10% of the ribosomes translate one specific codon, the concentration of ribosomes translating it is in the range 0.01–1 μM. In Fig. 2F,G, the slow codon has a ratio between the total number of tRNAs and total number of ribosomes of 1:4, and their concentration is 0.2 μM and 0.8 μM, respectively. For the fast codon, the ratio between the total number of tRNAs and total number of ribosomes is 4:1, and their concentration is 0.4 μM and 0.1 μM, respectively.

GTP concentration is found to be one order of magnitude below ATP concentration according to data from Koç et al (2004) and Takaine et al (2022). As an approximation, we assumed a GTP concentration proportional to ATP concentration in 1:10 ratio. Finally, the relative difference between the minimal and maximal ATP concentrations of the fast-to-slow TC loading rate ratio was computed using Eq. 6, for combinations of rate constants between $10^2$–$10^4$ for $k_{\text{ch}}$, $10^7$–$10^8$ for $k_{\text{l}}$, $10^4$–$10^6$ for $k_{\text{TC}}$, $10^5$–$10^7$ for $k_{\text{tr}}$, and $10^{-3}$–$10^{-1}$ for $k_{\text{TC rev}}$. For each interval, we considered all possible values in steps of 0.1 in the order of magnitude. The fast-to-slow

TC loading rate ratio was then computed for every parameter value combination (over 2.1 million combinations).

## Ischemic time analysis

The sample ischemic time in minutes is defined by GTEx as "the interval between actual death, presumed death, or cross clamp application and final tissue stabilization" and was obtained from the GTEx sample annotation file, column "SMTSISCH". The ischemic time was compared to the COMD coefficient adjusted for age and tissue, defined as the residual of the linear regression predicting the sample COMD coefficient as a linear combination of the age and a tissue effect.

## Yeast strains and culture conditions

The strain used in this study was BY4741 Mat a ura3 met1 his3 leu2 background (referred to as wild type). *S. cerevisiae* were grown in trehalose-containing medium as the non-fermentable carbon source at 30 °C. Trehalose-containing medium includes 20 g/l trehalose, 6.7 g/l YNB + $(\text{NH}_4)_2\text{SO}_4$ (yeast nitrogen base without amino acids; Difco), amino-acid supplements at a final concentration of 100 mg/l to complement the auxotrophies of the strains. The medium was buffered at pH 4.8 by adding 14.6 g/l succinic acid and 6 g/l NaOH (Walther, 2010). Pre-cultures were grown overnight in 250 mL flasks and agitated at 150 rpm. The next day, pre-cultures were diluted to $\text{OD}_{600} = 0.05$ and grown until an $\text{OD}_{600}$ of ~0.5 was reached. To block mitochondrial function, antimycin A was added to a final 2 μg/ml to the medium and incubated for 0, 2, 5, and 10 min (Walther, 2010). Cells were fast spun down for 15 s at $13,000 \times g$ in a microcentrifuge and flash-frozen in liquid nitrogen.

## HT-5PSeq library preparation

HT-5PSeq libraries were prepared as previously performed (Zhang and Pelechano, 2021). Briefly, 6 μg RNA was used for DNase treatment, then DNA-free total RNA were directly ligated with RNA/RNA oligo containing UMI (RNA rP5_RND oligo). Ligated RNA was reverse transcribed and primed with Illumina PE2 compatible oligos containing random hexamers and oligo-dT. RNA in RNA/DNA hybrid was depleted by sodium hydroxide within 20 min at 65 °C incubation. Ribosomal RNAs were depleted using DSN (Duplex-specific nuclease) with the mixture of ribosomal DNA probes. Samples were amplified by PCR and sequenced in an Illumina NextSeq 2000 instrument using 55 sequencing cycles for read 1 and 55 cycles for read 2.

## HT-5PSeq analysis

HT-5PSeq reads were trimmed 3′-sequencing adapter using cutadapt V1.16 (Martin, 2011). The 8-nt random barcodes on the 5′ ends of reads were extracted and added to the header of the fastq file as the UMI using UMI-tools. 5′P reads were mapped to the *S. cerevisiae* reference genome (SGD R64-1-1) using STAR 2.7.9a with the parameter --alignEndsType Extend5pOfRead1 to exclude soft-clipped bases on the 5′ end. After removing PCR duplicates by UMI-tools 1.0.0 (Smith et al, 2017), analysis of 5′ ends positions was performed using the Fivepseq package (Nersisyan et al, 2020), including the relative distance to start and stop codons. Specifically, the unique 5′mRNA reads in biological samples were merged. The

third replicate sample from the −5 min time point was discarded due to significantly lower 5PSeq coverage compared to other samples.

## 5PSeq modeling

We considered all 5PSeq reads located 17 nt 5′ of in-frame codons. To avoid possible confounding effects due to translation initiation and termination, we did not consider the start codon and the second codon, nor we considered the stop codon and its preceding codon. For robustness, we further did not consider genes with less than 10 reads mapping to all considered codons. Furthermore, genes encoded in mitochondrial DNA were discarded because they use a different genetic code.

We started by isolating the contribution of the codon-specific ribosome dwelling to the 5PSeq coverage from factors independent of translation elongation. The 5PSeq read coverage can depend on how frequently the corresponding mRNA is degraded, its expression and how frequently translation is initiated. Furthermore, as Xrn1 follows the last translating ribosome, the position of the codon in the coding sequence could potentially impact the 5PSeq coverage. Hence, we modeled the reads in gene $g$ and distance $d$ from the start codon in order to isolate the effect of the A-site codon on the 5PSeq reads from gene, position, and sequencing depth effects. For each sample (belonging to a batch and time point) we modeled the read coverage $y_{g,d}$ on gene $g$ at distance $d$ from the start codon (adjusting for the 17 nt shift) using a generalized linear model with a negative binomial distribution and the log link function:

$$y_{g,d} \sim NB(\mu_{g,d}, \theta)$$

$$\log(\mu_{g,d}) = \alpha_g + \beta_{k(g,d)} + \gamma d + s,$$

Where $\alpha_g$ is a gene effect, $\beta_{k(g,d)}$ is the codon-associated 5′ coverage for the A-site codon located at distance $d$ on gene $g$, and $s$ (or size factor) is a sample-specific intercept.

The model was fitted using the package *statsmodels* v0.12.0. Due to memory limitations, the model was fitted separately on three-thirds of the data for each sample, and the coefficient estimates averaged over the three-thirds.

To define optimal and non-optimal codons for the yeast 5PSseq time course, we considered the samples harvested prior to application of the drug. Next we averaged $\beta_k$ for every codon $k$ across these samples and defined the optimal codons as the codons from the first quartile and the non-optimal codons as the codons from the fourth quartile.

To relate intracellular ATP concentration with the 5P-seq readouts at time point 0 while accounting for centrifugation and sample stabilization of the 5PSeq protocol, we considered the intracellular ATP concentrations reported by (Walther, 2010) at 30 s. For the later time points (2 min, 5 min, 10 min), we used the very same time points of Walther.

## Cassette exons

Exon annotations and percent-spliced-ins (PSI) were obtained from the ASCOT annotation (Ling et al, 2020). We filtered for cassette exons according to ASCOT (i.e. feature cassette_exon=Yes) that belong to genes of the consensus CDS gene set of the GENCODE release 25 (build GRCh38). We further restricted the analysis to such cassette exons that fully overlap a coding sequence. This resulted in 29,417 cassette exons. The decoding rate of each coding cassette exon was then calculated as the geometric mean of the HEK293 decoding rates (Dana and Tuller, 2015) of the codons fully contained in the exon. A cassette exon was considered tissue-specifically differentially spliced if its PSI was at least 20 percent points above or below its average PSI across tissues.

## Data availability

The datasets and computer code produced in this study are available in the following databases: 5PSeq data: NCBI's Gene Expression Omnibus (Edgar et al, 2002) accessible through GEO Series accession number GSE216524. GTEx RNA-seq samples: Authorized researchers can access the BAM files for 7778 RNA-Seq samples, the gene-level and the transcript-level TPM (transcript per million) values, as well as the sample annotation of the GTEx v6 dataset from the GTEx portal. We are requesting permission from the GTEx project to share the exonic-to-intronic read count ratios, as these are computed from the non-publicly accessible BAM files. The BAM files of the Tabula Muris Senis atlas dataset (Almanzar et al, 2020) can be downloaded from the public AWS S3 bucket (https://registry.opendata.aws/tabula-muris-senis/). The code developed for the analysis is available at https://github.com/gagneurlab/Cellular_energy_codon_analysis. All other data are available in the manuscript or the supplementary materials (Presnyak et al, 2015; Roux and Topisirovic, 2012; Wu et al, 2019).

## Peer review information

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

## Acknowledgements

We would like to thank Diego Núñez, Thomas Becker, and Roland Beckmann for valuable advice and feedback on the manuscript. Furthermore, we are grateful to Alexander Karollus, Leonhard Wachutka, Felix Brechtmann and all the remaining members of the Gagneur Lab for valuable discussions. This study was supported by the Deutsche Forschungsgemeinschaft (DFG, German Research Foundation) - Project-ID 403584255 - TRR 267 (to ET and JG) and the German Bundesministerium für Bildung und Forschung (BMBF) through the Model Exchange for Regulatory Genomics project MERGE (031L0174A to JG). LDM is supported by the Helmholtz Association under the joint research school Munich School for Data Science - MUDS. PTS is funded by the Munich Center for Machine Learning (MCML). VP acknowledges the support from the Swedish Research Council (VR 2020-01480 and VR 2021-06112), a Wallenberg Academy Fellowship (KAW 2021.0167), the Swedish Foundations' Starting Grant (Ragnar Söderberg Foundation), Vinnova (2020-03620) and Karolinska Institutet (SciLifeLab Fellowship, SFO and KI funds). YZ is funded by a fellowship from the China Scholarship Council. The Genotype-Tissue Expression (GTEx) Project was supported by the Common Fund of the Office of the Director of the National Institutes of Health and by the National Cancer Institute, National Human Genome Research Institute, National Heart, Lung, and Blood Institute, National Institute on Drug Abuse, National Institute of Mental Health, and National Institute of Neurological Disorders and Stroke. The data used for the analyses described in this manuscript were obtained from the GTEx Portal on June 12, 2017, under accession number dbGaP: phs000424.v6.p1. Figures 1A, 3D and 5A as well as the synopsis image were created with Biorender.

## Author contributions

**Pedro Tomaz da Silva**: Conceptualization; Data curation; Software; Formal analysis; Investigation; Visualization; Methodology; Writing—original draft; Writing—review and editing. **Yujie Zhang**: Resources; Data curation; Software; Formal analysis; Investigation; Visualization; Methodology; Writing—review and editing. **Evangelos Theodorakis**: Data curation; Software; Formal analysis; Methodology. **Laura D Martens**: Data curation; Software; Methodology. **Vicente A Yépez**: Visualization; Project administration; Writing—review and editing. **Vicent Pelechano**: Conceptualization; Resources; Supervision; Funding acquisition; Methodology; Writing—review and editing. **Julien Gagneur**: Conceptualization; Supervision; Funding acquisition; Visualization; Methodology; Writing—original draft; Project administration; Writing—review and editing.

## Funding

## Disclosure and competing interests statement

The authors declare no competing interests.

