## [Peer Review File · Molecular Systems Biology]

Cellular energy regulates mRNA degradation in a codon-specific manner

Pedro Tomaz da Silva, Yujie Zhang, Evangelos Theodorakis, Laura Martens, Vicente Yépez, Vicent Pelechano, and Julien Gagneur

Corresponding author(s): Julien Gagneur (gagneur@in.tum.de)

Review Timeline:

Submission Date:	4th Jul 23
Editorial Decision:	16th Aug 23
Revision Received:	26th Oct 23
Editorial Decision:	1st Dec 23
Appeal Received:	24th Jan 24
Editorial Decision:	8th Feb 24
Revision Received:	19th Feb 24
Accepted:	20th Feb 24

Editor: Maria Polychronidou

Transaction Report:

16th Aug 2023

Manuscript Number: MSB-2023-11861

Title: Cellular energy regulates mRNA translation and degradation in a codon-specific manner

Dear Julien,

Thank you again for submitting your work to Molecular Systems Biology. We have now heard back from the three reviewers who agreed to evaluate your study. As you will see below the reviewers acknowledge that the addressed topic is relevant. They do however raise a series of concerns which we would ask you to address in a revision.

I think that the reviewers' comments seem clear and therefore I do not see the need to repeat them here. As reviewers #2 and #3 indicate, alternative explanations for the presented observations need to be examined and excluded or at minimum acknowledged. Overall, reviewers #2 and #3 raise some concerns regarding the conclusiveness of the study and provide suggestions on how to strengthen the work. All issues raised by the reviewers need to be satisfactorily addressed. As you may already know, our editorial policy allows in principle a single round of major revision, so it is essential to provide responses to the reviewers' comments that are as complete as possible. Please feel free to contact me in case you would like to discuss in further detail any of the issues raised. I would be happy to schedule a call.

On a more editorial level, we would ask you to address the following points:

- Please provide a .doc version of the manuscript text (including legends for the main figures and EV Figures) and individual production quality figure files for the main Figures and EV Figures (one file per figure).
- We have replaced Supplementary Information by the Expanded View (EV format). In this case, all additional figures can all be included in a PDF called Appendix. Appendix figures should be labeled and called out as: "Appendix Figure S1, Appendix Figure S2... Appendix Table S1..." etc. Each legend should be below the corresponding Figure/Table in the Appendix. Please include a Table of Contents in the beginning of the Appendix. For detailed instructions regarding expanded view please refer to our Author Guidelines: .
- Table EV1 is rather long and it should be provided as a separate file (xls or csv format). Please include a description of the Table in the file itself, either in a separate sheet (for xls file) or as a README.txt file zipped together with the table file (for csv file).
- Please include 5 keywords.
- Please provide a "standfirst text" summarizing the study in one or two sentences (approximately 250 characters), three to four "bullet points" highlighting the main findings and a "synopsis image" (550px width and max 400px height, jpeg format) to highlight the paper on our homepage.
- All Materials and Methods need to be described in the main text. We would encourage you to use 'Structured Methods', our new Materials and Methods format. According to this format, the Materials and Methods section should include a Reagents and Tools Table (listing key reagents, experimental models, software and relevant equipment and including their sources and relevant identifiers) followed by a Methods and Protocols section in which we encourage the authors to describe their methods using a step-by-step protocol format with bullet points, to facilitate the adoption of the methodologies across labs. More information on how to adhere to this format as well as downloadable templates (.doc or .xls) for the Reagents and Tools Table can be found in our author guidelines: . An example of a Method paper with Structured Methods can be found here:
- Please include a "Disclosure and Competing Interests Statement" in the main text.
- Please include a Data availability section describing how the data, code etc. have been made available. This section needs to be formatted according to the example below:
The datasets and computer code produced in this study are available in the following databases:
 - Chip-Seq data: Gene Expression Omnibus GSE46748 (<https://www.ncbi.nlm.nih.gov/geo/query/acc.cgi?acc=GSE46748>)
 - Modeling computer scripts: GitHub (<https://github.com/SysBioChalmers/GECKO/releases/tag/v1.0>)
 - [data type]: [full name of the resource] [accession number/identifier] ([doi or URL or identifiers.org/DATABASE:ACCESSION])
- For data quantification: please specify the name of the statistical test used to generate error bars and P values, the number (n) of independent experiments (specify technical or biological replicates) underlying each data point and the test used to calculate p-values in each figure legend. The figure legends should contain a basic description of n, P and the test applied. Graphs must include a description of the bars and the error bars (s.d., s.e.m.).

- Molecular Systems Biology supports formal data citations in the Reference list, to cite previously published datasets. In addition to citing the original papers that reported the data, we encourage you to also cite the relevant datasets directly in the Reference list. In the text, references to datasets are included as "Data ref: Smith et al, 2001" or "Data ref: NCBI Sequence Read Archive PRJNA342805, 2017". In the Reference list, data citations are very similar to normal literature references but must be labeled with "[DATASET]" at the end of the reference. For detailed instructions please refer to our Author Guidelines .

- The References should be formatted according to the Molecular Systems Biology reference style (i.e. ordered alphabetically and listing the first 10 authors followed by et al).

- When you resubmit your manuscript, please download our CHECKLIST (<https://bit.ly/EMBOPressAuthorChecklist>) and include the completed form in your submission.

Please note that the Author Checklist will be published alongside the paper as part of the transparent process (<https://www.embopress.org/page/journal/17444292/authorguide#transparentprocess>).

If you feel you can satisfactorily deal with these points and those listed by the referees, you may wish to submit a revised version of your manuscript. Please attach a covering letter giving details of the way in which you have handled each of the points raised by the referees. A revised manuscript will be once again subject to review and you probably understand that we can give you no guarantee at this stage that the eventual outcome will be favorable.

Kind regards,

Maria

Maria Polychronidou, PhD
Senior Editor
Molecular Systems Biology

We realize that it is difficult to revise to a specific deadline. In the interest of protecting the conceptual advance provided by the work, we recommend a revision within 3 months (14th Nov 2023). Please discuss the revision progress ahead of this time with the editor if you require more time to complete the revisions. Use the link below to submit your revision:

IMPORTANT: When you send your revision, we will require the following items:

1. the manuscript text in LaTeX, RTF or MS Word format
2. a letter with a detailed description of the changes made in response to the referees. Please specify clearly the exact places in the text (pages and paragraphs) where each change has been made in response to each specific comment given
3. three to four 'bullet points' highlighting the main findings of your study
4. a short 'blurb' text summarizing in two sentences the study (max. 250 characters)
5. a 'thumbnail image' (550px width and max 400px height, Illustrator, PowerPoint or jpeg format), which can be used as 'visual title' for the synopsis section of your paper.
6. Please include an author contributions statement after the Acknowledgements section (see <https://www.embopress.org/page/journal/17444292/authorguide>)
7. Please complete the CHECKLIST available at (<https://bit.ly/EMBOPressAuthorChecklist>). Please note that the Author Checklist will be published alongside the paper as part of the transparent process (<https://www.embopress.org/page/journal/17444292/authorguide#transparentprocess>).
8. When assembling figures, please refer to our figure preparation guideline in order to ensure proper formatting and readability in print as well as on screen:

See also figure legend guidelines: <https://www.embopress.org/page/journal/17444292/authorguide#figureformat>

9. Please note that corresponding authors are required to supply an ORCID ID for their name upon submission of a revised manuscript (EMBO Press signed a joint statement to encourage ORCID adoption).

(<https://www.embopress.org/page/journal/17444292/authorguide#editorialprocess>)

Currently, our records indicate that the ORCID for your account is 0000-0002-8924-8365.

Link Not Available

The system will prompt you to fill in your funding and payment information. This will allow Wiley to send you a quote for the article processing charge (APC) in case of acceptance. This quote takes into account any reduction or fee waivers that you may be eligible for. Authors do not need to pay any fees before their manuscript is accepted and transferred to the publisher.

EMBO Press participates in many Publish and Read agreements that allow authors to publish Open Access with reduced/no publication charges. Check your eligibility: <https://authorservices.wiley.com/author-resources/Journal-Authors/open-access/affiliation-policies-payments/index.html>

*** PLEASE NOTE *** As part of the EMBO Press transparent editorial process initiative (see our Editorial at <https://dx.doi.org/10.1038/msb.2010.72>), Molecular Systems Biology publishes online a Review Process File with each accepted manuscripts. This file will be published in conjunction with your paper and will include the anonymous referee reports, your point-by-point response and all pertinent correspondence relating to the manuscript. If you do NOT want this File to be published, please inform the editorial office at msb@embo.org within 14 days upon receipt of the present letter.

Reviewer #1:

In this manuscript da Silva and coworkers study the link between metabolism and mRNA stability mediated by mRNA translation. The authors observe that cellular energy regulates mRNA degradation in a codon-specific manner in mammals. Experiments to assay the short-term response of ribosome dwell time to intracellular ATP deprivation in yeast are consistent with ATP concentration modulating the effect of codon optimality on translation and translation-related mRNA degradation. Intriguingly, codon optimality of cassette exons increasingly included or skipped in a tissue relates to its codon optimality-mediated mRNA degradation coefficient.

The work is well done and the manuscript is of general interest addressing how codon optimality plays a role in differential gene expression independent of variations in the tRNA pool. Before publication one issue that should be addressed though it likely does not influence the overall findings and conclusion of the manuscript. To determine the COMD coefficient, the authors use HEK293 codon decoding rates determined in HEK293 cells by Dana & Tuller (2015) based on ribo-seq data by Lee et al. (2012). How consistent are decoding rates when compared to other HEK293 ribo-seq datasets and how different are these rates compared to e.g mouse ES cell data.

Reviewer #2:

In this manuscript, Silva et al. analyzed RNA-seq data sets from GTEx project that comprises thousands of post-mortem human samples. The authors probed variations of mRNA stability by computing changes of exonic and intronic reads. Using codon optimality-mediated mRNA degradation (COMD) coefficients, the authors found a correlation between ATP synthesis and COMD tendency. Based on 5P-seq using yeast cells, the authors concluded that cellular energy attenuates the effects of codon optimality via tRNA charging and loading.

This manuscript is more like a theory based on existing data sets. Although the point is attractive, the main conclusion lacks rigorous experimental validation. Varied ATP demand in different tissues could affect many fundamental aspects of mRNA translation besides tRNA charging and loading. Previous studies (PMID: 30029003) demonstrated that inhibiting nutrient signaling pathways alleviates amino acid shortage-mediated ribosome pausing. In the case of codon optimality, it is not surprising that non-optimal codons are more sensitive to energy availability. The authors should consider many other factors such as the initiation rate and degradation machinery that is also energy dependent. For the experimental validation, as the authors admit, yeast cells are not ideal. However, this reader is a bit surprised that the authors did not measure the mRNA stability in parallel with 5P-seq. Overall, this manuscript suffers from correlative analysis with too many overstatements.

Major concerns,

1. Page 4 "we considered foreach mRNA its decoding rate in HEK293 cells as a codon optimality measure (Dana & Tuller,2015)." The authors should validate their assumption that the codon decoding time is similar among human tissues. Dana and Tuller et al., 2014 used Ribo-seq data of HEK293 cells, so it would be possible to evaluate codon decoding time using mouse tissue Ribo-seq data using the same statistical model. Or the authors should make sure that the average ribosome densities of each codon are well correlated across tissues and not very different from those of HEK293 cells. It is certainly challenging to analyze tissue-specific Ribo-seq data, but the authors may at least clearly state their assumption and the possibilities of different decoding rates in the main text.

2. Page 9 "Ribosome occupancies profiling was performed using 5'P sequencing (5PSeq) (Pelechano et al, 2016; Zhang & Pelechano, 2021). Importantly, 5PSeq, which does not require in vitro incubation with RNase, allowed us to obtain measurements in the first few minutes after drug application." The authors used 5PSeq to validate the effects of ATP deprivation on ribosome occupancy in yeast. Since 5PSeq capture ribosome-protected footprints (RPFs) from mRNAs undergoing co-translational degradation, comparing the A-site codon occupancy obtained from 5PSeq between optimal and non-optimal codons may not correctly reflect the translational status of all the elongating ribosomes. This would make it difficult to interpret the result, even though the authors captured a clear ATP-dependent change in the difference of A-site ribosome occupancy between optimal and non-optimal codons. In my opinion, 5PSeq is not an alternative of Ribo-seq. To address the authors' question, Ribo-seq would be the best and the most straightforward experiment because it captures all the RPFs regardless of whether they are degradation intermediates or not.

3. Page 11 "our results show that cellular energy regulates the effect of codon optimality on mRNA stability and translation." The authors showed the mechanistic link between ATP levels and decoding rates using their mathematical modeling. But the ribosome pausing and stalling due to the slow decoding do not always cause mRNA degradation. Could the authors comment on whether ATP levels could also sensitize the ribosome pausing to degradation, or ATP levels just affect decoding rates?

Reviewer #3:

Codon usage bias and tRNA pools vary between tissues and cell types. This manuscript argues for another facet of this regulation—that the strength of codon-mediated mRNA decay (COMD), the accelerated turnover of mRNAs with disfavored codons, varies between tissues. Further, it is specifically claimed that the strength of COMD correlates with mitochondrial function, which varies between tissues, and that ATP levels mediate this effect.

The variation in mRNA degradation is assessed from large-scale RNA sequencing data, by comparing intronic reads (as a measure of mRNA synthesis) and exonic reads (as a measure of mRNA abundance) to extract mRNA degradation, the third parameter in a simple kinetic model. Inferred degradation across different transcripts is correlated with codon usage, recapitulating the COMD phenomenon in general, and the strength of this effect indeed varies between tissues. COMD strength is then found to correlate with gene expression signatures of mitochondrial function. A model is proposed wherein differences in ATP abundance and overall cell energy charge affects tRNA charging, leading to downstream effects on charged tRNA availability and decoding. This model is further tested in two ways. First, COMD effect is found to correlate with post-mortem ischemic time in RNA-seq samples, which is linked to altered cellular metabolism during this ischemic period. Second, distortions in ribosome occupancy are reported in budding yeast subjected to mitochondrial inhibition. Finally, alternatively spliced cassette exons are analyzed for exon-specific changes in effective COMD.

This manuscript argues for a subtle but important connection between cell physiology and mRNA decay, mediated by translation. The analysis of RNA-seq data supporting this concept appear robust, but certain important alternative explanations are not excluded, raising fundamental questions about the interpretation. Thus, while the data argue for some interesting change in mRNA metabolism across these different tissues, it is not clear how to link this causally with COMD. Notably, the kinetic model of translation neglects key features of actual aminoacylation that argue against the direct ATP effect proposed here.

1. mRNA degradation arises through COMD and through a range of other, codon-independent processes. If the activity of other mRNA decay pathways varies between cells, then the relative contribution of COMD and the strength of the correlation between half-life and codon usage would vary as well. That is, the analysis measures the fraction of half-life explained by COMD, and a change in this fraction could result from changes in other decay.

2. Likewise, increased persistence of intronic reads in certain tissues could compress the inferred decay rates and reduce the apparent correlation with codon usage.

3. Additionally, the mass-action kinetic model for translation does not capture important features of tRNA charging. Notably, these enzymes create adenylate-amino acid intermediates, and they are saturated for ATP at concentrations well below cellular levels. As a result, it seems unlikely that the fraction of charged tRNA is sensitive to ATP variation across the range seen in viable cells.

Prof. Dr. Julien Gagneur
Technische Universität München | Arcisstraße 21 | 80333 München

To: Maria Polychronidou
Editor, Molecular Systems Biology
Meyerhofstrasse 1
69117 Heidelberg
Germany

Munich, 26th of October 2023

Revised version of manuscript MSB-2023-11861

Dear Dr. Polychronidou,

We thank the reviewers for their constructive comments and their appraisal of the relevance of our study. We have implemented several important changes in order to address the main criticism that TC loading is only one among many possible mechanisms underlying our observation. Specifically:

- We have restructured the manuscript by moving the theoretical model to the end of the results section. This makes clear that the ischemic time analysis and the perturbation data in yeast test the link between cellular energy and codon effects, independently of the underlying responsible pathway(s). The theoretical model of TC loading is suggested at the end of the manuscript as one possible underlying pathway.
- We now call the theoretical model “the mathematical model” instead of “the model”. This is to make clear that the core observations hold independently of the role of TC loading.
- We now discuss the alternative mechanisms suggested by the reviewers. Importantly none of these are incompatible with a contribution of TC loading.
- We have included a new supplementary figure (Appendix Fig S4)

Toning down the TC loading model has improved the manuscript by giving more importance to the novel observations per se (association between codon effects and tissue, age, ischemic time, tissue-specific codon bias of cassette exons, and ribosome A-site residency upon ATP deprivation) and providing a less restrictive interpretation.

Moreover, we have realized that we forgot to mention that there are 3 biological replicates per time point the 5Pseq time series and that one sample of the -5 min data point was discarded due to aberrantly low coverage. This is now fixed.

We have furthermore addressed all other points in our point-by-point response below.

On behalf of all the authors,

Julien Gagneur

General response to all reviewers: We thank the reviewers for their insightful and constructive feedback. We have implemented several important changes in order to address the main criticism that TC loading is only one among many possible mechanisms underlying our observation. Specifically:

- We have restructured the manuscript by moving the theoretical model to the end of the results section. This makes clear that the ischemic time analysis and the perturbation data in yeast test the link between cellular energy and codon effects, independently of the underlying responsible pathway(s). The theoretical model of TC loading is suggested at the end of the manuscript as one possible underlying pathway.

- We now call the theoretical model “the mathematical model” instead of “the model”. This is to make clear that the core observations hold independently of the role of TC loading.

- We now discuss the alternative mechanisms suggested by the reviewers. Importantly none of these are incompatible with a contribution of TC loading.

- We have included a new supplementary figure (Appendix Fig S4)

Toning down the TC loading model has improved the manuscript by giving more importance to the novel observations per se (association between codon effects and tissue, age, ischemic time, tissue-specific codon bias of cassette exons, and ribosome A-site residency upon ATP deprivation) and providing a less restrictive interpretation.

Moreover, we have realized that we forgot to mention that there are 3 biological replicates per time point the 5Pseq time series and that one sample of the -5 min data point was discarded due to aberrantly low coverage. This is now fixed.

Reviewer #1:

In this manuscript da Silva and coworkers study the link between metabolism and mRNA stability mediated by mRNA translation. The authors observe that cellular energy regulates mRNA degradation in a codon-specific manner in mammals. Experiments to assay the short-term response of ribosome dwell time to intracellular ATP deprivation in yeast are consistent with ATP concentration modulating the effect of codon optimality on translation and translation-related mRNA degradation. Intriguingly, codon optimality of cassette exons increasingly included or skipped in a tissue relates to its codon optimality-mediated mRNA degradation coefficient.

The work is well done and the manuscript is of general interest addressing how codon optimality plays a role in differential gene expression independent of variations in the tRNA pool. Before publication one issue that should be addressed though it likely does not influence the overall findings and conclusion of the manuscript. To determine the COMD coefficient, the authors use HEK293 codon decoding rates determined in HEK293 cells by Dana & Tuller (2015) based on ribo-seq data by Lee et al. (2012). How consistent are decoding rates when compared to other HEK293 ribo-seq datasets and how different are these rates compared to e.g mouse ES cell data.

Response:

Unfortunately, Dana and Tuller do not provide the code to compute codon decoding rates from Ribo-Seq. Since the procedure they use is hard to replicate we opted for addressing this point by comparing the decoding rates they estimated on different Ribo-Seq datasets including mouse and human. However, we had shown that human decoding rates estimated in Dana and Tuller highly correlate with other metrics of codon optimality such as CSC (codon stabilization coefficient, Supplementary figure 3).

Moreover, we have now compared the HEK293 decoding rates to mouse Embryonic stem cells (Spearman's $\rho = 0.54$ $p < 1e-5$, see below) and to mouse Neutrophils ($\rho = 0.62$ $p < 1e-6$, see below). These results show that mouse and human codon decoding rates are similar to each other despite coming from different species, cell lines, Ribo-Seq datasets and laboratories.

Importantly, computing COMD in human using decoding rates derived from mouse embryonic stem cells and in mouse neutrophils results in a very similar ranking of tissues in their COMD coefficient. This indicates that the overall trend is independent of the Ribo-Seq dataset considered to obtain a measure for decoding rate. We have included these new results in the manuscript (page 4, paragraph 2 last sentence and new Appendix Fig S4).

Reviewer #2:

In this manuscript, Silva et al. analyzed RNA-seq data sets from the GTEx project that comprises thousands of post-mortem human samples. The authors probed variations of mRNA stability by computing changes of exonic and intronic reads. Using codon optimality-mediated mRNA degradation (COMD) coefficients, the authors found a correlation between ATP synthesis and COMD tendency. Based on 5P-seq using yeast cells, the authors concluded that cellular energy attenuates the effects of codon optimality via tRNA charging and loading.

Response: Our manuscript may have given a wrong impression to this reviewer. While our data shows that ATP deprivation affects codon-specific ribosome occupancy experimentally in yeast, the model of tRNA charging is proposed as one possible pathway but it is not directly assayed. We have reworked the relevant statements to make this clearer as well as restructured the manuscript to emphasize that the theoretical model is one possible interpretation.

Reviewer #2: This manuscript is more like a theory based on existing data sets.

Response: We are not entirely clear what the reviewer wants to express with this statement. Theories are helpful in science, and leveraging existing data too. The mammalian transcriptome data were published before but our observations are novel (association between codon effects and tissue, age, ischemic time, tissue-specific codon bias of cassette exons). Furthermore, we did generate new perturbation experiments in yeast. Perhaps, the reviewer is concerned that the tRNA loading aspect of the paper was not assayed directly. We hope that the restructuring and rewriting of the manuscript makes this clearer.

Reviewer #2: Although the point is attractive, the main conclusion lacks rigorous experimental validation. Varied ATP demand in different tissues could affect many fundamental aspects of mRNA translation besides tRNA charging and loading. Previous studies (PMID: 30029003) demonstrated that inhibiting nutrient signaling pathways alleviates amino acid shortage-mediated ribosome pausing.

Response: We are now discussing alternative models to tRNA loading for the dependency mechanisms that would depend both on energy availability and codon optimality that we establish.

The study by M. Darnell et al (PMID: 30029003) focuses on the sensitivity of ribosome pausing to deprivation of two amino acids, leucine or arginine. The authors find that upon arginine deprivation, ribosome pausing occurs for some arginine codons consistently across 3 cell lines of study. In contrast, upon leucine deprivation, ribosome pausing is not consistently observed on leucine codons across their cell lines of study. Ribosome pausing in specific arginine codons associates with the percentage of charged cognate tRNAs. The authors further found that the absence of a response of the GCN2 or mTORC1 pathways during leucine or arginine limitation is sufficient to deplete charged tRNA pools and induce genome-wide ribosome pausing at cognate codons.

It is possible that the mechanism behind their observations is at place in some of our analysed tissues, complementary to other mechanisms. Nevertheless, our results indicate a global modulation of codon optimality effects on mRNA half-life rather than a regulation at the level of individual codons. We are now referring to this manuscript and mention the possibility for companion ATP-sensing and signaling mechanisms regulating translation in the discussion (page 13, paragraph 3).

Reviewer #2: In the case of codon optimality, it is not surprising that non-optimal codons are more sensitive to energy availability. The authors should consider many other factors such as the initiation rate and degradation machinery that is also energy dependent.

Response: As this reviewer states, it might not be surprising in retrospect that non-optimal codons are more sensitive to energy availability than optimal ones. However, we are the first to report this association, to demonstrate it experimentally, and to investigate its implication in tissue-specific regulation and during aging. We agree that tRNA loading is not the only possible pathway that could explain our observations. We have now expanded the discussion to mention more possibilities including ATP-dependent changes in initiation rate and degradation machinery, and how they connect with our findings (page 13, paragraph 3).

Reviewer #2: For the experimental validation, as the authors admit, yeast cells are not ideal. However, this reader is a bit surprised that the authors did not measure the mRNA stability in parallel with 5P-seq.

Response: We have not performed mRNA stability experiments such as SLAM-Seq due to the transient change in the ATP pools. Our ribosome position data and ATP information shows that the phenotype is very dynamic and changes under 2 minutes. Furthermore, changes in ribosome occupancy under the short time period may not immediately lead to measurable changes in mRNA stability. We have now included this point in the discussion (page 13, paragraph 1).

Reviewer #2: Overall, this manuscript suffers from correlative analysis with too many overstatements.

Response: We hope that the rewriting addresses the overstatements mentioned by the reviewer. We refer to the overview of the changes in the response to the editor above.

Reviewer #2: Major concerns,

1. Page 4 "we considered foreach mRNA its decoding rate in HEK293 cells as a codon optimality measure (Dana & Tuller,2015)." The authors should validate their assumption that the codon decoding time is similar among human tissues. Dana and Tuller et al., 2014 used Ribo-seq data of HEK293 cells, so it would be possible to evaluate codon decoding time using mouse tissue Ribo-seq data using the same statistical model. Or the authors should make sure that the average ribosome densities of each codon are well correlated across tissues and not very different from

those of HEK293 cells. It is certainly challenging to analyze tissue-specific Ribo-seq data, but the authors may at least clearly state their assumption and the possibilities of different decoding rates in the main text.

Response: We do not assume in the manuscript that the codon decoding time is similar among human tissues (or under ATP changes). In fact Fig 1B suggests the opposite, as well as the yeast ribosome occupancy data in yeast. Instead, we used the decoding rate in HEK293 as a measure of optimality that overall correlates with the association pattern we are observing in Fig 1B. We have expanded the results section related to Fig1B-C to clarify this (page 4, paragraph 1). We have also investigated alternative reference decoding rates in response to reviewer 1 (new Appendix Fig 4).

Reviewer #2: 2. Page 9 "Ribosome occupancies profiling was performed using 5'P sequencing (5PSeq) (Pelechano et al, 2016; Zhang & Pelechano, 2021). Importantly, 5PSeq, which does not require in vitro incubation with RNase, allowed us to obtain measurements in the first few minutes after drug application." The authors used 5PSeq to validate the effects of ATP deprivation on ribosome occupancy in yeast. Since 5PSeq capture ribosome-protected footprints (RPFs) from mRNAs undergoing co-translational degradation, comparing the A-site codon occupancy obtained from 5PSeq between optimal and non-optimal codons may not correctly reflect the translational status of all the elongating ribosomes. This would make it difficult to interpret the result, even though the authors captured a clear ATP-dependent change in the difference of A-site ribosome occupancy between optimal and non-optimal codons. In my opinion, 5PSeq is not an alternative of Ribo-seq. To address the authors' question, Ribo-seq would be the best and the most straightforward experiment because it captures all the RPFs regardless of whether they are degradation intermediates or not.

Response: Despite being very powerful, the limitation of ribosome profiling is that it requires a long *in vitro* incubation in presence of RNases. Although that is valid in many instances, the sample handling required for ribosome profiling makes it almost impossible to obtain clean results in under 2 minutes. In contrast, 5PSeq leverages the intrinsic toeprinting activity of RNases obtained in the cell, without the need of polysome fractionation, in vitro RNA digestion, or the other extensive manipulations. Therefore, it can be performed at very short times by using flash freezing followed by sequencing.

Moreover, we agree with the reviewer that 5PSeq and ribosome profiling measure different subpools of ribosomes. It is not clear why codon occupancies would differ among RNAs being co-translationally degraded and intact RNAs. If there were a difference though, focusing on the co-translationally degraded RNAs would actually be an advantage as it narrows down the mRNAs where COMD is more likely to have been triggered.

We have now included a section in the discussion addressing these two points (page 12, paragraph 4).

Reviewer #2: 3. Page 11 "our results show that cellular energy regulates the effect of codon optimality on mRNA stability and translation." The authors showed the mechanistic link between ATP levels and decoding rates using their mathematical modeling. But the ribosome pausing and stalling due to the slow decoding do not always cause mRNA degradation. Could the authors comment on whether ATP levels could also sensitize the ribosome pausing to degradation, or ATP levels just affect decoding rates?

Response: We rewrote the discussion to integrate more alternative possible mechanisms including this relevant suggestion. We cannot exclude that higher ATP levels could sensitize the ribosome pausing to degradation, making this step less dependent on codon identity. However, this would not explain the changes of codon occupancies that we observed in yeast. Hence, this possible and complementary mechanism is not sufficient to explain all our observations (page 13, paragraph 3).

Reviewer #3:

Codon usage bias and tRNA pools vary between tissues and cell types. This manuscript argues for another facet of this regulation-that the strength of codon-mediated mRNA decay (COMD), the accelerated turnover of mRNAs with disfavored codons, varies between tissues. Further, it is specifically claimed that the strength of COMD correlates with mitochondrial function, which varies between tissues, and that ATP levels mediate this effect.

The variation in mRNA degradation is assessed from large-scale RNA sequencing data, by comparing intronic reads (as a measure of mRNA synthesis) and exonic reads (as a measure of mRNA abundance) to extract mRNA degradation, the third parameter in a simple kinetic model. Inferred degradation across different transcripts is correlated with codon usage, recapitulating the COMD phenomenon in general, and the strength of this effect indeed varies between tissues. COMD strength is then found to correlate with gene expression signatures of mitochondrial function. A model is proposed wherein differences in ATP abundance and overall cell energy charge affects tRNA charging, leading to downstream effects on charged tRNA availability and decoding. This model is further tested in two ways. First, COMD effect is found to correlate with post-mortem ischemic time in RNA-seq samples, which is linked to altered cellular metabolism during this ischemic period. Second, distortions in ribosome occupancy are reported in budding yeast subjected to mitochondrial inhibition. Finally, alternatively spliced cassette exons are analyzed for exon-specific changes in effective COMD.

This manuscript argues for a subtle but important connection between cell physiology and mRNA decay, mediated by translation. The analysis of RNA-seq data supporting this concept appear robust, but certain important alternative explanations are not excluded, raising fundamental questions about the interpretation. Thus, while the data argue for some interesting change in mRNA metabolism across these

different tissues, it is not clear how to link this causally with COMD. Notably, the kinetic model of translation neglects key features of actual aminoacylation that argue against the direct ATP effect proposed here.

Reviewer #3: 1. mRNA degradation arises through COMD and through a range of other, codon-independent processes. If the activity of other mRNA decay pathways varies between cells, then the relative contribution of COMD and the strength of the correlation between half-life and codon usage would vary as well. That is, the analysis measures the fraction of half-life explained by COMD, and a change in this fraction could result from changes in other decay.

Response: We rewrote the discussion to integrate more alternative possible mechanisms including this relevant possible mechanism. Indeed, we cannot exclude that higher ATP levels could promote codon-independent degradation pathways, making mRNA degradation less dependent on codon identity. However, this would not explain the changes of codon occupancies that we observed in yeast. Thus, this possible and complementary mechanism is not sufficient to explain all our observations. We have now integrated this point in the discussion (page 13, paragraph 3).

Reviewer #3: 2. Likewise, increased persistence of intronic reads in certain tissues could compress the inferred decay rates and reduce the apparent correlation with codon usage.

Response: A global change in intronic read persistence is possible. However, this would not explain the ribosome A-site residency variations we observe in yeast. Moreover, our codon association heatmaps (fig 1B) and the COMD are linear regression coefficients. Assuming that a tissue specific persistence of intronic reads could be modeled as a tissue-specific multiplicative factor common to all genes, this will lead to a change in the intercept of our regression of the log-transformed exonic-to-intronic read count ratio but would not affect the regression coefficients. More complex relationships between a tissue and its intronic read persistence could be at play although it isn't clear how they could take shape. We are now including this possibility in the discussion (page 13, paragraph 1).

Reviewer #3: 3. Additionally, the mass-action kinetic model for translation does not capture important features of tRNA charging. Notably, these enzymes create adenylate-amino acid intermediates, and they are saturated for ATP at concentrations well below cellular levels. As a result, it seems unlikely that the fraction of charged tRNA is sensitive to ATP variation across the range seen in viable cells.

Response: First, the mathematical model is simple and proposes a possible explanation, however it is not central to our main finding and report. We have restructured and reworded parts of the manuscript to make this clearer. Second, we have now investigated an alternative model where the tRNA charging is at saturated ATP levels as this reviewer suggests. To this end, we have simplified the tRNA aminoacylation reaction rate to be

$V_{tRNA \text{ aminoacylation}} = k_{ch}[tRNA]_{free}$ similarly to what has been modeled for instance by Guimaraes et al. 2020. Genome Biology.

The results can be seen below. We find that for an extensive representative range of rate constants the loading ratio decrease follows a very similar distribution to the model without ATP saturation, perhaps consistent with the fact these enzymes function at saturated levels. Note that ATP abundance still influences the systems via its conversion to GTP.

We did not find a reference regarding these enzymes being saturated for ATP. We would appreciate a reference this reviewer may have in order to motivate and include these analyses in a revised version.

Editorial points:

- Please provide a .doc version of the manuscript text (including legends for the main figures and EV Figures) and individual production quality figure files for the main Figures and EV Figures (one file per figure). ✓

- We have replaced Supplementary Information by the Expanded View (EV format). In this case, all additional figures can all be included in a PDF called Appendix. Appendix figures should be labeled and called out as: "Appendix Figure S1, Appendix Figure S2... Appendix Table S1..." etc. Each legend should be below the corresponding Figure/Table in the Appendix. Please include a Table of Contents in the beginning of the Appendix. For detailed instructions regarding expanded view please refer to our Author Guidelines: <http://msb.embopress.org/authorguide#expandedview>. ✓

- Table EV1 is rather long and it should be provided as a separate file (xls or csv format). Please include a description of the Table in the file itself, either in a separate sheet (for xls file) or as a README.txt file zipped together with the table file (for csv file). ✓

- Please include 5 keywords.

codon optimality, mRNA stability, mRNA translation, cellular energy metabolism, tissue-specific regulation

- Please provide a "standfirst text" summarizing the study in one or two sentences (approximately 250 characters), three to four "bullet points" highlighting the main findings and a "synopsis image" (550px width and max 400px height, jpeg format) to highlight the paper on our homepage.

Standfirst text: Analysis of GTEx data and perturbation experiments in yeast show that cellular energy regulates the effect of codon optimality on mRNA stability and translation

Bullet Points:

- Codon optimality matters more in conditions of scarcer energy, such as tissues with low mitochondrial activity, older age, oxygen deprivation, or exposure to specific drugs.
- This effect can explain up to 2-fold variation in mRNA stability between human tissues and is reflected in the codon usage of tissue-specific cassette exons
- Validation experiments corroborated by mathematical modeling show that codon decoding kinetics respond nonuniformly to ATP abundance independently of tRNA regulation

- All Materials and Methods need to be described in the main text. We would encourage you to use 'Structured Methods', our new Materials and Methods format. According to this format, the Materials and Methods section should include a Reagents and Tools Table (listing key reagents, experimental models, software and relevant equipment and including their sources and relevant identifiers) followed by a Methods and Protocols section in which we encourage the authors to describe their methods using a step-by-step protocol format with bullet points, to facilitate the adoption of the methodologies across labs. More information on how to adhere to this format as well as downloadable templates (.doc or .xls)

for the Reagents and Tools Table can be found in our author guidelines: <https://www.embopress.org/page/journal/17444292/authorguide#textformat> . An example of a Method paper with Structured Methods can be found here: <https://www.embopress.org/doi/10.15252/msb.20178071>. This does not apply to this paper. Our manuscript refers to Zhang and Pelechano 2021 for the 5Pseq experiment, the only wet lab experiment of our study. This publication in Cell rep. methods provides a detailed protocol.

- Please include a "Disclosure and Competing Interests Statement" in the main text. ✓

- Please include a Data availability section describing how the data, code etc. have been made available. This section needs to be formatted according to the example below:

The datasets and computer code produced in this study are available in the following databases:

- Chip-Seq data: Gene Expression Omnibus GSE46748
(<https://www.ncbi.nlm.nih.gov/geo/query/acc.cgi?acc=GSE46748>)

- Modeling computer scripts: GitHub
(<https://github.com/SysBioChalmers/GECKO/releases/tag/v1.0>)

- [data type]: [full name of the resource] [accession number/identifier] ([doi or URL or identifiers.org/DATABASE:ACCESSION]) ✓

- For data quantification: please specify the name of the statistical test used to generate error bars and P values, the number (n) of independent experiments (specify technical or biological replicates) underlying each data point and the test used to calculate p-values in each figure legend. The figure legends should contain a basic description of n, P and the test applied. Graphs must include a description of the bars and the error bars (s.d., s.e.m.). ✓
We included a clarification that the error bar in figure 3F belongs to standard deviation from a permutation test.

- Molecular Systems Biology supports formal data citations in the Reference list, to cite previously published datasets. In addition to citing the original papers that reported the data, we encourage you to also cite the relevant datasets directly in the Reference list. In the text, references to datasets are included as "Data ref: Smith et al, 2001" or "Data ref: NCBI Sequence Read Archive PRJNA342805, 2017". In the Reference list, data citations are very similar to normal literature references but must be labeled with "[DATASET]" at the end of the reference. For detailed instructions please refer to our Author Guidelines <http://msb.embopress.org/authorguide#datacitation>. Does not apply.

- The References should be formatted according to the Molecular Systems Biology reference style (i.e. ordered alphabetically and listing the first 10 authors followed by et al). ✓

- When you resubmit your manuscript, please download our CHECKLIST (<https://bit.ly/EMBOPressAuthorChecklist>) and include the completed form in your submission.

Please note that the Author Checklist will be published alongside the paper as part of the transparent process (<https://www.embopress.org/page/journal/17444292/authorguide#transparentprocess>). ✓

1st Dec 2023

RE: Manuscript MSB-2023-11861R, Cellular energy regulates mRNA translation and degradation in a codon-specific manner

Dear Julien,

Thank you again for submitting your revised manuscript to Molecular Systems Biology. We invited reviewers #2 and #3 to evaluate your revised study. Unfortunately, reviewer #2 was not available this time. We have now heard back from reviewer #3. As you will see below, reviewer #3 still raises substantial concerns on your work, which unfortunately preclude its publication in Molecular Systems Biology.

Overall, the reviewer is still not convinced that the reported conclusions are well supported and raises issues regarding the analysis and interpretation of the data. They rated the conclusiveness of the study as "Low" and indicated that they do not support publication in Molecular Systems Biology. Given the substantial remaining concerns in combination with the fact that our editorial policy allows in principle a single round of major revision, I am afraid that we cannot offer to publish the study.

While we cannot pursue this manuscript further, we encourage you to transfer your study to our not-for-profit open-access sister journal, Life Science Alliance (LSA). We shared your manuscript and the accompanying reviews with LSA Executive Editor, Eric Sawey, who is interested in these findings, and would like to invite further consideration of this manuscript at LSA, revised to acknowledge the limitations outlined by Reviewer 3.

We encourage you to use the link below to transfer your manuscript to LSA. You do not need to revise the manuscript before transferring it to LSA. Once you transfer, Dr. Sawey will email you an invitation to revise and resubmit, listing the same revision requests as mentioned above. Please feel free to reach out at e.sawey@life-science-alliance.org if you have any questions about the LSA journal, the transfer process or the revisions requested.

I apologize for not bringing better news regarding the publication of your study in Molecular Systems Biology and I hope you will view the possibility of a transfer to Life Science Alliance favorably.

Kind regards,

Maria

Maria Polychronidou, PhD
Senior Editor
Molecular Systems Biology

Reviewer #3:

Revisions have not addressed my concerns with the interpretations presented in the data. Overall, the manuscript describes well-designed analyses of data that argue for differential impact of COMD across different cell types and conditions.

My sense is that these trends can be explained by shifts in the relative strength of COMD and translation-independent pathways that alter the fraction of all mRNA decay that goes through COMD versus other, translation-independent pathways.

The budding yeast experiments are not immune from this concern. It is claimed that,

"Another advantage of 5PSeq over Ribo-Seq is that 5PSeq only profiles ribosome occupancies of co-translationally degraded mRNAs, therefore focusing on mRNAs for which COMD is more likely to have been triggered."

For the current analysis, however, this is a confounding factor and a disadvantage. Ribo-Seq measurements would provide a degradation-independent test of A-site occupancy, whereas 5Pseq is affected by changes in degradation. Notably, the "long in vitro incubation times" in Ribo-Seq occur after freezing and lysing cells, and rapid phenomena have been studied with this method.

As a more specific but important point, Figure 3C appears to depict experimental measurements of ATP levels, but as far as I can tell, these are meant to represent something conceptual and there is no direct evidence that ATP levels follow this trajectory.

The figure should be re-worked so it's clear that it is conceptual (or if these are real data then this should be clarified and the methods used to make these measurements should be described).

I also remain concerned that tRNA charging probably does not follow mass-action kinetics. There is quite a bit of enzymology carried out on tRNA synthetases, for instance see doi:10.1016/j.ymeth.2007.09.007 and many citations therein. Importantly, the kinetic schemes of synthetases lead to weak dependence of charging rate on ATP concentration, a qualitative difference from mass-action kinetics that would raise difficulties for the proposed model.

** As a service to authors, EMBO Press offers the possibility to directly transfer declined manuscripts to another EMBO Press title or to the open access journal Life Science Alliance launched in partnership between EMBO Press, Rockefeller University Press and Cold Spring Harbor Laboratory Press. The full manuscript and if applicable, reviewers' reports, are automatically sent to the receiving journal to allow for fast handling and a prompt decision on your manuscript. For more details of this service, and to transfer your manuscript please click on Link Not Available. **

Response to the 2nd reviews

Reviewer 3: “Revisions have not addressed my concerns with the interpretations presented in the data. Overall, the manuscript describes well-designed analyses of data that argue for differential impact of COMD across different cell types and conditions.

My sense is that these trends can be explained by shifts in the relative strength of COMD and translation-independent pathways that alter the fraction of all mRNA decay that goes through COMD versus other, translation-independent pathways.”

Response: We are glad to read that this reviewer notes that our “manuscript describes well-designed analyses of data that argue for differential impact of COMD across different cell types and conditions.” Reviewer 3 further raises a valid concern regarding the potential underlying mechanism, which had already been raised in the first revision. In response to this concern, we had already included a section in the discussion (page 13, paragraph 3).

To make the reader more aware of this limitation and of the possibility of translation-independent mechanisms, we have now implemented further changes, namely:

- we have now changed the manuscript title from “Cellular energy regulates mRNA translation and degradation in a codon-specific manner” to “Cellular energy regulates mRNA degradation in a codon-specific manner”. With this, we make clear that our main claims and observations come from mRNA degradation and leave open the possibility for translation to be involved.

- we have now changed the description of results and legends in Figure 3 to incorporate the fact that changes in codon-associated 5' fragments in 5PSeq data might not be exclusively driven by ribosome occupancy changes. Notably we now denote the metric “codon-associated 5' coverage“ instead of “ribosome A-site residency”.
- We expanded the discussion section about translation-independent pathways to incorporate how it could potentially explain the changes in the 5PSeq signal we observe.

We believe that even if this alternative mechanism holds, our observations are important and novel. We identify for the first time a modulation of the contribution of codon optimality to mRNA degradation dependent on cellular energy, although we leave room for further research on the exact underlying mechanism. All the rest being the same, if upon ATP deprivation, translation-independent pathways are relatively less used than translation-dependent pathways including COMD, then the relative contribution of COMD to RNA degradation will be higher, and therefore the importance of codon optimality would be amplified. We cannot exclude this mechanism. Whether it is taking place or not does not invalidate claims 1-4.

Reviewer 3: “The budding yeast experiments are not immune from this concern. It is claimed that,

‘Another advantage of 5PSeq over Ribo-Seq is that 5PSeq only profiles ribosome occupancies of co-translationally degraded mRNAs, therefore focusing on mRNAs for which COMD is more likely to have been triggered.’

For the current analysis, however, this is a confounding factor and a disadvantage. Ribo-Seq measurements would provide a degradation-independent test of A-site occupancy, whereas 5Pseq is affected by changes in degradation.

Response: RiboSeq data may help distinguish between a shift in the translation-independent degradation versus COMD by measuring A-site protection independently of mRNA degradation. Independently of the underpinning molecular mechanism, we consider the number of novel results (Claims 1-4), including the perturbation data provided in Fig 3 that present not only 5PSeq in ATP-synthesis-inhibited yeast but also mRNA stability measures in humans as a function of ischemic times, to be substantial enough to warrant publication and trigger future research.

Reviewer 3: Notably, the ‘long in vitro incubation times’ in Ribo-Seq occur after freezing and lysing cells, and rapid phenomena have been studied with this method.”

Response: As for the study of rapid phenomena, 5PSeq is capable of detecting alterations occurring within just 1 minute. This is due to the fact that 5PSeq can be executed with just 1 mL of sample volume, enabling the utilization of Eppendorf-based collection methods. In contrast, ribosome profiling necessitates larger volumes (typically over 200 mL), thereby extending the

collection time. Such quick responsiveness is particularly valuable when observing the immediate effects of drug treatments on ATP concentrations. We have now included this point into the discussion, while acknowledging the limitations of 5PSeq. We have also rewritten the section to make more clear the limitations of the methods we used.

Reviewer 3: “As a more specific but important point, Figure 3C appears to depict experimental measurements of ATP levels, but as far as I can tell, these are meant to represent something conceptual and there is no direct evidence that ATP levels follow this trajectory. The figure should be re-worked so it’s clear that it is conceptual (or if these are real data then this should be clarified and the methods used to make these measurements should be described).”

Response: The lack of labels on Figure 3C y-axis had probably confused this reviewer. Our manuscript was mentioning both in the methods and figure 3F that we take intracellular ATP concentration as reported by Walther et al, 2010. Our 5PSeq experiment was performed exactly on the same conditions. Figure 3C is not a conceptual cartoon but is generated from this data. We have now added y-axis labels to makes this clear.

Reviewer 3: “I also remain concerned that tRNA charging probably does not follow mass-action kinetics. There is quite a bit of enzymology carried out on tRNA synthetases, for instance see doi:10.1016/j.ymeth.2007.09.007 and many citations therein. Importantly, the kinetic schemes of synthetases lead to weak dependence of charging rate on ATP concentration, a qualitative difference from mass-action kinetics that would raise difficulties for the proposed model.”

Response: We have responded to this point in the last revision. Our response to this point was two-fold. First, we restructured the manuscript and made it clear that the main claims (1-4) are independent of the mathematical model. Second, we have performed simulations assuming independence of the tRNA synthetase reaction to ATP levels, hence deviating from mass action kinetics for this step. The conclusions remain qualitatively the same and quantitatively very similar.

Ultimately, our data supports the effect of cellular energy on COMD, whether by affecting codon decoding rates or by altering the relative strength of COMD versus translation-independent pathways. Consequently, deeming the conclusiveness of our study as “low” seems to us unjustified. Moreover, basing the rejection of our paper solely on the feedback of one reviewer, who did not provide a detailed, point-by-point analysis and seems not fully aware of the revisions we made, does not constitute a solid foundation for rejecting our manuscript.

In light of these considerations, we respectfully request a re-evaluation of our manuscript.

8th Feb 2024

Manuscript Number: MSB-2023-11861RR-Q

Title: Cellular energy regulates mRNA translation and degradation in a codon-specific manner

Dear Julien,

Thank you for your message asking us to reconsider our decision on your manuscript MSB-2023-11861R. I have now had the chance to evaluate the points raised in your appeal letter and I have also consulted with reviewer #1, who had evaluated the initial version of the study. (This reviewer had not been asked to evaluate the revised version, since they had not raised major issues on the initial version.) As you will see below, reviewer #1 thinks that your answers to the remaining concerns of reviewer #3 seems reasonable. Considering the toned-down conclusions and more balanced interpretation of the presented findings, reviewer #1 is overall supportive of publication and only recommends some additional discussion to broaden the scope of the conclusions. You can see the specific comments of reviewer #1 below. Taken together, we would ask you to perform the changes suggested by reviewer #1 in a minor revision. We would also ask you to address some editorial issues listed below.

- Our data editors have noted the following information that needs to be added/corrected in the Figure legends:
 - The box plots need to be defined in terms of minima, maxima, centre, bounds of box and whiskers, and percentile in the legends of figures 3e; 4a.
 - The information related to n is missing in the legends of figures 3e-f.
- The funding information provided in the manuscript text (Acknowledgements) should match the information entered in the online submission system. Currently the following information is missing from the submission system: "The Common Fund of the Office of the Director of the National Institutes of Health and by the National Cancer Institute, National Human Genome Research Institute, National Heart, Lung, and Blood Institute, National Institute on Drug Abuse, National Institute of Mental Health, and National Institute of Neurological Disorders and Stroke".
- Please include 5 keywords.
- Please remove the 'Authors Contributions' from the manuscript. The 'Author Contributions' section is replaced by the CRediT contributor roles taxonomy to specify the contributions of each author in the journal submission system. Please use the free text box in the 'author information' section of the online submission system to provide more detailed descriptions if needed (e.g., 'X provided intracellular Ca⁺⁺ measurements in fig Y').
- Appendix Table S1 should be provided as Dataset EV1. Please include a description of the dataset in a separate sheet in the xls. The callout in the text should be corrected accordingly.
- The Figure legends should be listed below the References.
- The Supporting Information section should be removed from the main text.
- Please provide a "standfirst text" summarizing the study in one or two sentences (approximately 250 characters), three to four "bullet points" highlighting the main findings to highlight the paper on our homepage.

Please resubmit your revised manuscript online, with a covering letter listing amendments and responses to each point raised by the referees. Please resubmit the paper ****within one month**** and ideally as soon as possible. If we do not receive the revised manuscript within this time period, the file might be closed and any subsequent resubmission would be treated as a new manuscript. Please use the Manuscript Number (above) in all correspondence.

Click on the link below to submit your revised paper.

Kind regards,

Maria

Maria Polychronidou, PhD
Senior Editor
Molecular Systems Biology

Reviewer #1

I have read the manuscript again and the comments and responses by the authors. In my opinion the authors argue and address appropriately all issues except "the Ribo-Seq measurements would provide a degradation-independent test of A-site occupancy". Here I agree with reviewer #3. Since the authors now rephrase their conclusion to "Cellular energy regulates mRNA degradation in a codon-specific manner", I think the ribo-seq would be nice but is not really required.

The mass-action issue seems to be resolved by the authors performing simulations assuming independence of the tRNA synthetase reaction to ATP levels, hence deviating from mass action kinetics for this step. The outcome here is that the conclusions remain qualitatively the same and quantitatively very similar.

Overall, in my opinion the authors have performed a solid set of analyses to support their now softened conclusions. These findings will most certainly be widely discussed and the conclusions are thought provoking and likely be contested. The coupling of energy metabolism with the regulation of gene expression is in itself very interesting. We know that nuclear transcription influences decay (PMID:22466169, 34272303). Cell size does so (37944513). The phase distribution of cell during the cell cycle influences energy metabolism and decay (24489117, 10.1101/2024.01.11.575159). All these effect are connected and translation and codon usage likely play a role, too. As the authors nicely state: Overall, these potential mechanisms linking cellular energy to codon optimality could take place to varying degrees and are not mutually exclusive. Identifying which mechanisms take place and delineating their contribution requires further research. Perhaps the authors can add a few thoughts to their discussion to broaden the scope.

If you do choose to resubmit, please click on the link below to submit the revision online before 9th Mar 2024.

IMPORTANT: Please note that corresponding authors are required to supply an ORCID ID for their name upon submission of a revised manuscript (EMBO Press signed a joint statement to encourage ORCID adoption).

(<https://www.embopress.org/page/journal/17444292/authorguide#editorialprocess>)

Currently, our records indicate that the ORCID for your account is 0000-0002-8924-8365.

Link Not Available

*** PLEASE NOTE *** As part of the EMBO Press transparent editorial process initiative (see our Editorial at <https://dx.doi.org/10.1038/msb.2010.72> , Molecular Systems Biology will publish online a Review Process File to accompany accepted manuscripts. When preparing your letter of response, please be aware that in the event of acceptance, your cover letter/point-by-point document will be included as part of this File, which will be available to the scientific community. More information about this initiative is available in our Instructions to Authors. If you have any questions about this initiative, please contact the editorial office (msb@embo.org).

All editorial and formatting issues were resolved by the authors.

20th Feb 2024

Manuscript number: MSB-2023-11861RRR

Title: Cellular energy regulates mRNA degradation in a codon-specific manner

Dear Julien,

Thank you again for sending us your revised manuscript. We are now satisfied with the modifications made and I am pleased to inform you that your paper has been accepted for publication.

Kind regards,

Maria

Maria Polychronidou, PhD
Senior Editor
Molecular Systems Biology
